

# Iterative multilinear optimization for planar model fitting under geometric constraints

Jorge Azorin-Lopez[1], Marc Sebban[2], Andres Fuster-Guillo[1], Marcelo Saval-Calvo[1] and Amaury Habrard[2]

[1] Department of Computer Technology, University of Alicante, Alicante, Spain
[2] Laboratoire Hubert-Curien, Université Jean Monnet, Saint Etienne, France

## ABSTRACT

Planes are the core geometric models present everywhere in the three-dimensional real world. There are many examples of manual constructions based on planar patches: facades, corridors, packages, boxes, etc. In these constructions, planar patches must satisfy orthogonal constraints by design (*e.g.* walls with a ceiling and floor). The hypothesis is that by exploiting orthogonality constraints when possible in the scene, we can perform a reconstruction from a set of points captured by 3D cameras with high accuracy and a low response time. We introduce a method that can iteratively fit a planar model in the presence of noise according to three main steps: a clustering-based unsupervised step that builds pre-clusters from the set of (noisy) points; a linear regression-based supervised step that optimizes a set of planes from the clusters; a reassignment step that challenges the members of the current clusters in a way that minimizes the residuals of the linear predictors. The main contribution is that the method can simultaneously fit different planes in a point cloud providing a good accuracy/speed trade-off even in the presence of noise and outliers, with a smaller processing time compared with previous methods. An extensive experimental study on synthetic data is conducted to compare our method with the most current and representative methods. The quantitative results provide indisputable evidence that our method can generate very accurate models faster than baseline methods. Moreover, two case studies for reconstructing planar-based objects using a Kinect sensor are presented to provide qualitative evidence of the efficiency of our method in real applications.

## INTRODUCTION

Fitting multiple plane-based models under geometric constraints to point clouds obtained with RGBD noisy sensors in time-constrained applications remains a challenging problem. More broadly, geometric primitive detection (planes, cuboids, spheres, cylinders, etc.) has been extensively studied for multiple applications (robotics, modeling, shape processing, rendering, interaction, animation, architecture, etc.) (*Kaiser, Ybanez Zepeda & Boubekeur, 2019*). Multiple plane-based primitives are particularly interesting because they are very common in several engineering and architectonic projects in industry and construction, configuring objects and scenarios designed with geometric characteristics

Corresponding author
Jorge Azorin-Lopez,
jazorin@dtic.ua.es

between planes (angles, position, etc.). Building corridors, facades or rooms, manufactured components, packages, boxes, etc., are commonly formed by planar patches. These engineering and architectonic elements are present in different application domains, such as robot navigation, object reconstruction and reverse engineering (*Werghi et al., 1999*; *Benko et al., 2002*; *Anwer & Mathieu, 2016*).

Computer vision plays an important role in providing methods for modeling objects or scenes by means of processing 3D point clouds. Plane detection and model fitting are frequently used as the first stage in object and scene modeling pipelines. Robot navigation systems use plane detection and fitting to perform Simultaneous Localization and Mapping (SLAM) tasks (*Lu & Song, 2015*; *Xiao et al., 2011*). Indoor and outdoor scene reconstruction may be performed with plane detection and fitting phases followed by setting up the piecewise planar model (*Cohen et al., 2016*; *Zhang et al., 2015*; *Engelmann & Leibe, 2016*). Object detection and reconstruction from 3D point clouds is a complex task in the computer vision field, frequently involving a prior plane detection and fitting step (*Hu & Bai, 2017*, *Saval-Calvo et al., 2015b*, *Saval-Calvo et al., 2015a*).

Various devices (*Sansoni, Trebeschi & Docchio, 2009*) can be used to obtain three-dimensional (3D) point clouds depending on the application such as medical imaging, industry and robotics. Recently, 3D consumer RGBD cameras have been used extensively in video games and other areas. Point clouds from RGBD cameras are commonly noisy owing to the presence of outliers and missing data (*Villena-Martínez et al., 2017*). The challenge of model fitting this type of point cloud forces the method to be robust against noise. However, different application areas, such as SLAM or Human–Computer Interatcion (HCI), introduce real-time constraints that must be satisfied by the system.

In this context, we propose a novel iterative method to accurately reconstruct scenes composed by planes from noisy 3D point clouds, using the geometric constraints of the scene. The main contribution of this method is that it can simultaneously fit different planes in a point cloud using linear regression estimators and simple constraints (normal vectors to the planes), providing a robust solution. It exhibits high accuracy in the presence of noise and outliers and reduces the processing time.

The reminder of this paper is organized as follows. First, we review studies related to multi-plane model reconstruction focusing on methods that exploit orthogonality constraints to optimize the relationship between accuracy and temporal cost. In "MC-LSE: An Iterative Multi-Constraint Least Squares Estimation Algorithm", we present the method, MC-LSE, for the multi-constraint least squares estimation (MC-LSE). In "Experiments", we describe an extensive experimental study of the method compared with other related methods. Finally, the conclusions of the experimental analysis are presented in "Discussion and Conclusions".

## Related work

The multi-class multi-model fitting is a classic problem dealing with the interpretation of a set of input points originating from noisy observations as a multiple model with different geometric primitives (*Barath & Matas, 2019*). Common cases of this problem are the estimation of multiple line segments and circles in 2D edge maps, homographies from

point correspondences, and multiple motions in videos, planes and spheres in 3D point clouds.

A particular case of model fitting is planar model fitting, which estimates a model composed of a set of planes that are represented by a noisy 3D point cloud. Problems such as point cloud segmentation or clustering are aimed at grouping points in subsets with similar characteristics (geometric, radiometric, etc.), often with unsupervised processes, providing plane detection (*Grilli, Menna & Remondino, 2017*), without addressing model fitting. Several authors categorize model fitting methods into three groups: Hough transform-based methods, iterative methods (regression and RANSAC), and region-growing-based methods (*Kaiser, Ybanez Zepeda & Boubekeur, 2019*; *Xie, Tian & Zhu, 2020*; *Jin et al., 2017*; *Deschaud & Goulette, 2010*). This common classification includes methods not only related to model fitting but also for segmentation and clustering in plane detection.

Region-growing methods build regions by expanding the area from seeds as certain conditions are met. A plane detection algorithm was proposed in (*Poppinga et al., 2008*) using a two-point-seed-growing approach. Different clustering methods have been proposed, such as k-means (*Cohen-Steiner, Alliez & Desbrun, 2004*), which is one of the most commonly used methods. In some cases, these methods are focused on clustering or region segmentation (*Nurunnabi, Belton & West, 2012*). The success of these methods usually depends on the selection of the initial seed. Furthermore, they are oriented toward plane detection rather than a complete model fitting process.

Hough transform (*Hough, 1962*) is a common method for parameterized object detection, such as lines or circles, formerly defined for 2D images (*Duda & Hart, 1972*). Extensions to deal with 3D images have been proposed in several studies (*Hulik et al., 2014*). The voting process of Hough transform-based methods results in a high computational cost, especially in the presence of large input data. To reduce the computational cost, the randomized Hough transform (RHT) satisfies the voting process in a probabilistic manner (*Borrmann et al., 2011*). The RHT is adequate for detecting planes in large structures. However, these methods have not yet demonstrated their efficiency in complex model-fitting problems of scenes with geometric constraints between planes.

Random sample consensus (RANSAC) is a robust approach for fitting single models in an iterative manner (*Fischler & Bolles, 1981*). The method randomly selects an initial subset of the data to calculate a tentative model. It is then validated by counting the remaining data whose distance is under a threshold (*i.e.* inliers). The process iterates, and the best model is finally selected. In computer vision, RANSAC and its variants are widely used owing to their robustness against noise in the input data. Some methods aimed at fitting planes in images or 3D data are based on RANSAC. In some cases, the least squares estimation (LSE) and RANSAC are used together as the LSE method is used to calculate the model from the initial subset. However, RANSAC tends to simplify complex planar structures (*Jin et al., 2017*). To overcome this problem, several variants have been proposed (*Saval-Calvo et al., 2015a*). The CC-RANSAC (*Gallo, Manduchi & Rafii, 2011*) and its variants (*Zhou et al., 2011*; *Zhou et al., 2013*) can obtain multiple surfaces by employing a

modified RANSAC loss function to obtain better results. These approaches select multiple subsets (one per expected plane) and consider the relationships among them, instead of selecting one random data and trying to find the best planar model by counting the inliers. CC-RANSAC includes the largest connected component of inliers with eight-neighbor topology. The CC-RANSAC variants improve the basic method by adding vector normal information to allow the estimation of each cluster in the clustering and patch-joining steps. However, these methods do not consider the information related to the planes themselves. MC-RANSAC (*Saval-Calvo et al., 2015a*) uses a pre-clustering process by using a k-means algorithm to estimate the clusters and a search tree technique to improve the solutions while considering the prior constraints (angles between planes). Moreover, it extends traditional RANSAC by introducing a novel step for evaluating whether the inliers comply with the prior constraints among pre-clusters. Hence, this method outperforms the previous methods, achieving high accuracy in the final model estimation. However, the introduction of the search tree to calculate the pre-clusters and the step in RANSAC to check the constraints among plane models make the method too slow for some applications. New methods have been developed based on RANSAC. Prior-MLESAC (*Zhao et al., 2020*) is based on the previous maximum likelihood estimation sampling consensus (MLESAC), which improves the extraction of vertical and non-vertical planar and cylindrical structures by exploiting prior knowledge of physical characteristics. Progressive-X (Prog-X) is an any-time algorithm for geometric multi-model fitting using the termination criterion adopted from RANSAC, improving similar methods in terms of accuracy (*Barath & Matas, 2019*; *Barath et al., 2020*).

Regression, which is one of the most referred approaches for plane fitting, is a statistical strategy to solve the problem by finding a model that minimizes the overall error in the data. Ordinary LSE and its variants such as least median of squares (LMS) regression are widely used methods. The least squares method is a standard approach used to estimate model parameters by minimizing the squared distances between the observed data and their expected values. In computer vision, it has been widely used as the most popular form of regression analysis in various tasks. However, LSE results are highly influenced by outliers, leading to inconsistent results (*Mitra & Nguyen, 2003*). LMS (*Massart et al., 1986*) minimizes the median of the squares and has proved to be more robust than the original LSE. However, it still fails when more than 50% of the data are outliers. Although other robust approaches have been proposed to overcome this problem, such as the least K-th order of square (LKS) or adaptive LKS (*Lee, Meer & Park, 1998*), the estimation of the optimal parameters requires high computational effort. Consequently, they are not viable for many applications, as the original LSE is one of the most used methods for this purpose, at least as a part of more sophisticated systems. LSE has been widely used to estimate planes or planar patches in computer vision or as a part of robust methods that can calculate them. *Araújo & Oliveira (2020)* provided a new robust statistical approach, robust statistic plane detection (RSPD), for detecting planes in unorganized point clouds to achieve better accuracy and response times than previous approaches.

Regression, which is one of the most referred approaches for plane fitting, is a statistical strategy to solve the problem by finding a model that minimizes the overall error in the data. Ordinary LSE and its variants such as least median of squares (LMS) regression are widely used methods. The least squares method is a standard approach used to estimate model parameters by minimizing the squared distances between the observed data and their expected values. In computer vision, it has been widely used as the most popular form of regression analysis in various tasks. However, LSE results are highly influenced by outliers, leading to inconsistent results (*Mitra & Nguyen, 2003*). LMS (*Massart et al., 1986*) minimizes the median of the squares and has proved to be more robust than the original LSE. However, it still fails when more than 50% of the data are outliers. Although other robust approaches have been proposed to overcome this problem, such as the least K-th order of square (LKS) or adaptive LKS (*Lee, Meer & Park, 1998*), the estimation of the optimal parameters requires high computational effort. Consequently, they are not viable for many applications, as the original LSE is one of the most used methods for this purpose, at least as a part of more sophisticated systems. LSE has been widely used to estimate planes or planar patches in computer vision or as a part of robust methods that can calculate them. *Araújo & Oliveira (2020)* provided a new robust statistical approach, robust statistic plane detection (RSPD), for detecting planes in unorganized point clouds to achieve better accuracy and response times than previous approaches.

In recent studies, the problem of multi-model fitting have been addressed using energy functions to balance geometric errors (*Isack & Boykov, 2012*) and solve multiple geometric models, where greedy approaches such as RANSAC do not perform properly. These global energy-based approaches search for an optimal solution to fit all models present in the multi-structured data set, usually at high computational costs with respect to the number of models fitted (*Amayo et al., 2018*). Moreover, these methods (PEARL, T-linkage and CORAL (*Isack & Boykov, 2012*; *Magri & Fusiello, 2014*; *Amayo et al., 2018*)) have shown their efficiency in solving 2D multi-model fitting problems and homographies, but do not show extensive experimentation in the reconstruction of 3D multiple plane-based models with geometric constraints. *Lin et al. (2020)* formulated the problem as a global gradient minimization, proposing an updated method (Global-L0) based on a constraint model that outperforms traditional plane fitting methods.

A review of related works shows that methods for model fitting considering both accuracy and computational cost are needed. We propose a method for plane-based models reconstructed from a set of 3D points, taking advantage of the geometric constraints that are present in the original scene, exhibiting high accuracy in the presence of noise and outliers, and reducing the processing time. We selected the most representative methods for comparison: LSE (*Mitra & Nguyen, 2003*) and RANSAC (*Fischler & Bolles, 1981*) as classic baseline methods and MC-RANSAC (*Saval-Calvo et al., 2015a*), RSPD (*Araújo & Oliveira, 2020*), Prior-MLESAC (*Zhao et al., 2020*), Prog-X (*Barath & Matas, 2019*; *Barath et al., 2020*) and Global-L0 (*Lin et al., 2020*) are the most recent, providing a wide range of methods.

# MC-LSE: AN ITERATIVE MULTI-CONSTRAINT LEAST SQUARES ESTIMATION ALGORITHM

In this section, we present our iterative method named MC-LSE for multi-constraint least-squares estimation. After the introduction of our notations in "Notations and Setting", we present the workflow of MC-LSE in "MC-LSE", and it involves three main steps: clustering, linear regression, and reassignment.

## Notations and setting

Let us assume that we have access to a set of $m$ points, $P = \{p_i = (p_i^x, p_i^y, p_i^z) \in \mathbb{R}^3\}_{i=1}^m$, captured by using a 3D camera and assumed to be (likely noisy) representatives of some target planar model (*e.g.* cube, facade, corridor and office). The task we are dealing with in this study consists of reconstructing this target model by learning $n$ planes under orthogonality constraints. These constraints take the form of an $n \times n$-matrix, $A$, where $A[j, k] = 1$ if planes $j$ and $k$ must be orthogonal, and 0 otherwise. In this supervised machine learning setting, $(p_i^x, p_i^y)$ plays the role of the input feature vector, and $p_i^z$ is the dependent variable (corresponding to the depth). We approach this task by solving a joint constrained regression problem (which is described in the next section), where the minimizer takes the form of a set of $n$ models, $h_{\theta_j}(p_i^x, p_i^y)$ (with $j = 1\ldots n$), that maps linearly from input $(p_i^x, p_i^y)$ to output $p_i^z$. $h_{\theta_j}(p_i^x, p_i^y)$ is supposed to provide a good estimation of $\hat{p}_i^z$ of $p_i^z$ as follows:

$$\theta_j^0 + \theta_j^1 p_i^x + \theta_j^2 p_i^y + \theta_j^3 \hat{p}_i^z = 0.$$

Therefore, we deduce that $h_{\theta_j}(p_i^x, p_i^y)$ is defined as follows:

$$h_{\theta_j}(p_i^x, p_i^y) = \frac{\theta_j^0 + \theta_j^1 p_i^x + \theta_j^2 p_i^y}{-\theta_j^3},$$

where $h_{\theta_j}$ is the $j^{th}$ plane, and $\theta_j = \left(\theta_j^0, \theta_j^1, \theta_j^2, \theta_j^3\right)$ is the corresponding set of parameters learned from a certain subset of points, $P_j \subset P$.

Let $N_j = (\theta_j^1, \theta_j^2, -\theta_j^3)$ be the normal vector of the $j^{th}$ plane and $\tilde{N}_l$ be the normal vector of any set of points $P_l \subset P$. $\tilde{N}_l$ can be easily computed by selecting the eigenvector corresponding to the smallest eigenvalue of the scatter matrix of $P_l$. Finally, let $kNN(p_i) \subset P$ be the set of $k$-nearest neighbors of $p_i$, given a metric distance.

## MC-LSE

MC-LSE is an iterative algorithm that aims at reconstructing the target model from a set of points captured by using a 3D camera. The workflow of our method is presented in Fig. 1, where the target model is a cube. It involves three main steps.

1. A **clustering-based unsupervised step** that consists of initializing $n$ clusters from set $P$ of points that are supposed to represent at most the $n$ planes viewed by the 3D camera. Note that this step is performed only once.

2. A **linear regression-based supervised step** that learns, under orthogonal constraints (using matrix $A$), $n$ planes from the clusters.

3. A **reassignment step** that challenges the membership of the points to the current clusters in a manner that minimizes the residuals of the regression tasks.

Steps 2 and 3 are repeated until no (or only a few) reassignments are performed.

### Clustering step

As illustrated in Fig. 2, performing clustering from set $P$ might be a tricky task. The points captured by the 3D camera (i) are often noisy representatives of the actual faces of the cube and (ii) may represent overlapping areas, especially at the corners of the cube (Fig. 2 (left)). Therefore, using a standard clustering algorithm with the Euclidean distance (such as $K$-Means (*Forgy, 1965*) as used in this study) would lead to irrelevant clusters that would tend to bring together points (see $p_1$ and $p_2$ in the figure) that likely do not belong to the same plane. To overcome this drawback, an efficient solution consists of projecting a 3D-point, $p_i$, onto a 6-dimensional space by considering not only the original features, $(p_i^x, p_i^y, p_i^z)$, but also the 3D-normal vector, $\tilde{N}_{kNN(p_i)}$, of set $kNN(p_i)$ of the nearest neighbors of $p_i$ (see Fig. 2 (right)). Let $\tilde{p}_i \in \mathbb{R}^6$ be the corresponding point. Set $\tilde{P} = \{\tilde{p}_i \in \mathbb{R}^6\}_{i=1}^m$ is then used as the input for the clustering algorithm that outputs $n$ clusters, $P_1,\ldots,P_n$. Note that if the number of points assigned to a given cluster is not large enough (according to a threshold tuned by the user), the corresponding plane is (at least partially) hidden and cannot be properly captured by the camera. In such a case, the cluster is deleted, and only $n-1$ planes are learned. Furthermore, note that the size of the neighborhood, $k$ (which is tuned by cross-validation), affects the homogeneity of the orientation of the normal vectors. The normal is smoothed for a large neighborhood, but the edges of the objects are also smoothed; thus, they are less descriptive. In contrast, if $k$ is small, the normal vectors are significantly affected by noise and less uniform for a single plane surface. To consider the specificity of the application at hand (quality of the 3D camera, level of noise in the data, and camera point of view), we suggest assigning a weight to the normal vector, which is also tuned by cross-validation.

The next step of the process is optimization under the constraints of parameters $\theta_1,\ldots,\theta_n$ corresponding to the $n$ planes of the target model. Note that even if the clustering step considers the normal vectors to prevent the algorithm from building irrelevant clusters, outliers may still have a considerable impact on the slope of the learned planes at the first iteration of MC-LSE (see Fig. 3A in the case of a target cube). To address this limitation, we suggest selecting landmarks as a certain percentage of points from each cluster, $P_i$, that are the closest (according to the Manhattan distance) to the centroid of $P_i$. Thus, the initialization of our planes should be improved (see Fig. 3B).

### Regression step

The regression step aims to use the points of the current clusters to fit parameters $\theta_1,\ldots,\theta_n$ of the 3D planes such that the orthogonality constraints of matrix $A$ are satisfied. To achieve this task, only the original coordinates of the points $(p^x, p^y, p^z)$ are used. Planes $h_{\theta j}$ ($j = 1\ldots n$) are of the following form:

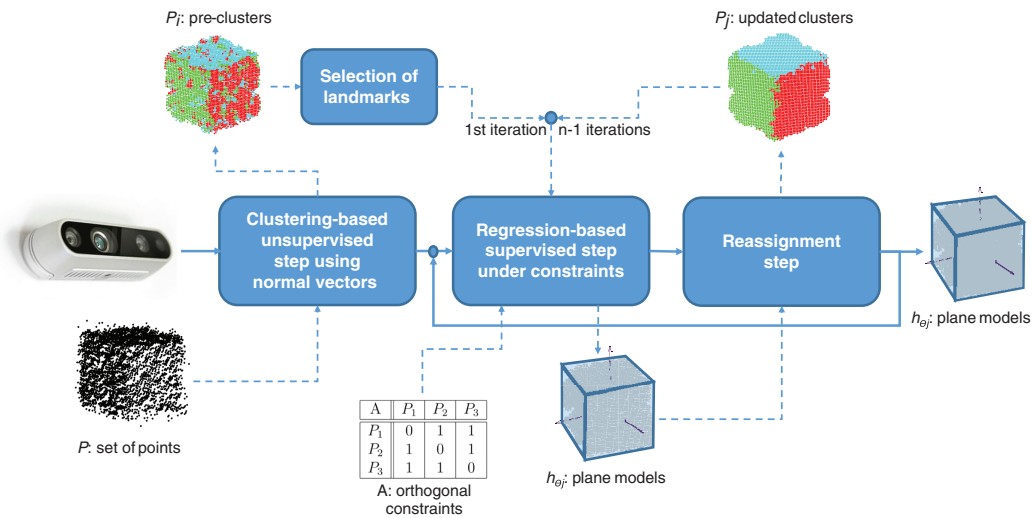

**Figure 1 Workflow of MC-LSE involving three main steps.**

$$h_{\theta_j}(p_i^x, p_i^y) = \frac{\theta_j^0 + \theta_j^1 p^x + \theta_j^2 p^y}{-\theta_j^3}.$$

The corresponding normal vector of each plane is given by $N_j = \left(\theta_j^1, \theta_j^2, -\theta_j^3\right)$. To correctly reconstruct the target planar model, $h_{\theta 1},...,h_{\theta n}$ must be learned under constraints such that the normal vectors, $N_i, N_j$, are orthogonal; that is, $N_i^t N_j = 0$—if $A[i, j] = 1$. Note that these constraints can be rewritten in terms of $\theta_i$ and $\theta_j$. Let $L$ be a $4 \times 4$-matrix defined as follows:

$$L = \begin{bmatrix} 0 & 0 & 0 & 0 \\ 0 & 1 & 0 & 0 \\ 0 & 0 & 1 & 0 \\ 0 & 0 & 0 & 1 \end{bmatrix}.$$

We can deduce that $N_i^t N_j$ can be rewritten as follows:

$$N_i^t N_j = \theta_i L \theta_j. \tag{1}$$

Planes $h_{\theta j}(p)$ ($j = 1...n$) is learned to find the $4 \times n$ matrix $\theta = (\theta_1,...,\theta_n)$ as the minimizer of the following constrained optimization least-squares problem.

$$\begin{aligned} \min_{\theta} \quad & 3l \sum_{i=1}^{n} \left( \frac{(\theta_i^0, \theta_i^1, \theta_i^2)^T X_i}{-\theta_i^3} - Z_i \right)^2 \\ s.t. \quad & \theta_i^T L \theta_j = 0 \qquad\qquad \text{if } A[i,j] = 1 \\ & 0 < \theta_i^3 \leq 1 \end{aligned} \tag{2}$$

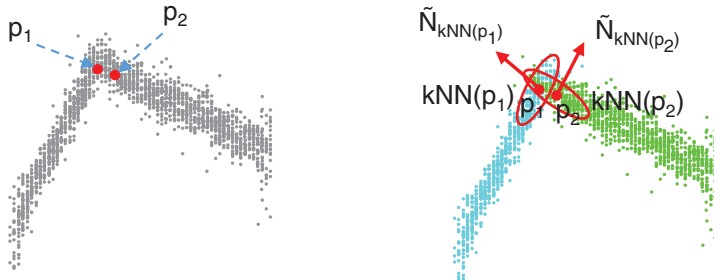

**Figure 2 Construction of the pre-clusters using the normal vectors of the points.** For simplicity, only two faces are considered here. On the left: a set of noisy points captured by the 3D camera. Points $p_1$ and $p_2$ are very close to each other according to the Euclidean distance in $\mathbb{R}^3$. On the right: a 6-dimensional feature vector is used to represent each point. The three additional features correspond to the normal 3D-vector, $\tilde{N}_{kNN(p_i)}$, calculated from the neighborhood, $kNN(p_i)$, of each point, $p_i$. In this way, $p_1$ and $p_2$ are no longer close and will probably belong to two different clusters, $C_1$ and $C_2$, shown in blue and green, respectively.

Here,

- $X_i = \left\{ \left(1, p_k^x, p_k^y\right) \right\}_{k=1}^{|P_i|}$ is the set of training examples of the $i^{th}$ current cluster, $P_i$, used to learn the $i^{th}$ plane.
- $Z_i = \left\{ p_k^z \right\}_{k=1}^{|P_i|}$ is the set of dependent values.

### Reassignment step

The objective of the reassignment is to challenge the membership of the points to the clusters, allowing us to better fit, using a step-by-step method, the parameters of the planes. Similar to the clustering step, the reassignment of $p_j$, assumed to be currently part of cluster $P_l$, accounts for both the original features, $(p_j^x, p_j^y, p_j^z)$, and the 3D-normal vector, $\tilde{N}_{kNN(p_j)}$, to prevent two close examples belonging to two different faces from being reassigned to the same cluster. The only difference is that the normal vector, $\tilde{N}_{kNN(p_j)}$, is calculated from the nearest neighbors that belong to $P_l$ instead of from the entire dataset, $P$. Let $\tilde{p}_j$ be the projection of $p_j$ in $\mathbb{R}^6$.

The reassignment of every point, $\tilde{p}_j$, consists of looking for the closest plane, $h_{\theta i}$, in terms of the Euclidean distance and assigning it to the corresponding cluster. In other words, this procedure selects a plane that minimizes the residuals. As illustrated in Fig. 4, this procedure allows us to significantly improve the quality of the clusters, having a positive impact on the planes optimized at the next iteration.

## EXPERIMENTS

In this section, we present an extensive experimental study of the proposed algorithm, MC-LSE. After the presentation of the experimental setup, we report the results of comparisons with previous methods in terms of several evaluation criteria. Then, we present an analysis of MC-LSE according to different levels of noise added to the data. The results were used for the cases presented in the next section.

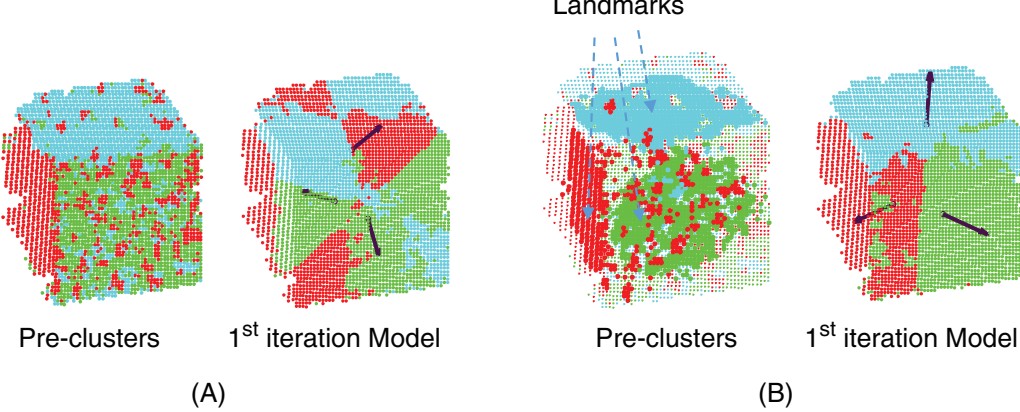

Landmarks

Pre-clusters    1st iteration Model        Pre-clusters    1st iteration Model

(A)                                (B)

**Figure 3  Selection of landmarks to improve the initialization of the iterative procedure.** (A) Normal vectors of the three visible faces (black arrows) that would be obtained from the planes generated from the whole pre-clusters. Because of the presence of noise, the quality of the reconstructed target model (here, a cube) is low. (B) The quality of the induced normal vectors is much higher by selecting landmarks from each cluster.               

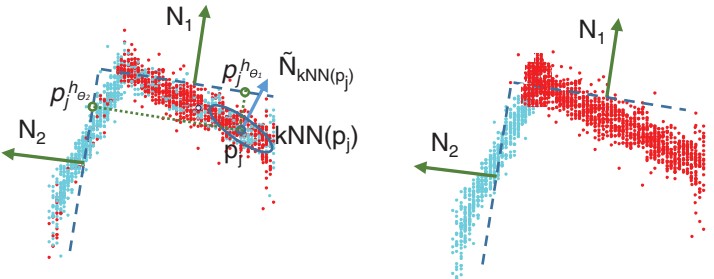

**Figure 4  Illustration of the reassignment step: distance calculation between the clusters and the estimated planes,** $h_{\theta_i}$ **(left); reassignment according to the minimum distance (right).**

## Experimental setup

Synthetic data from a target cube were used for the experiments. Without loss of generality, this setup allows us to quantitatively compare the methods on a simple target model by having access to the ground truth (analytical expression of the true planes of the cube). In this cube-based scenario, the number of learned planar models was set to $n = 3$, and matrix $A$ was based on 2-by-2 orthogonality constraints, as presented in Table 1.

Synthetic data were generated by simulating a Microsoft Kinect sensor with Blensor (*Gschwandtner et al., 2011*). This tool allows us to create a cube and obtain images from different points of view by moving a virtual camera. The experiments included eight points of view of the cube (see Fig. 5), simulating a counter-clockwise self-rotation of the object to validate the algorithm in terms of different geometrical and clusters characteristics. The method was implemented with MATLAB 2019 and YALMIP, a toolbox for modeling and optimization (*Löfberg, 2004*).

**Table 1 Target cube and the corresponding matrix of constraints *A*.**

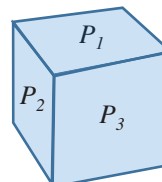

| A | $P_1$ | $P_2$ | $P_3$ |
|---|---|---|---|
| $P_1$ | 0 | 1 | 1 |
| $P_2$ | 1 | 0 | 1 |
| $P_3$ | 1 | 1 | 0 |

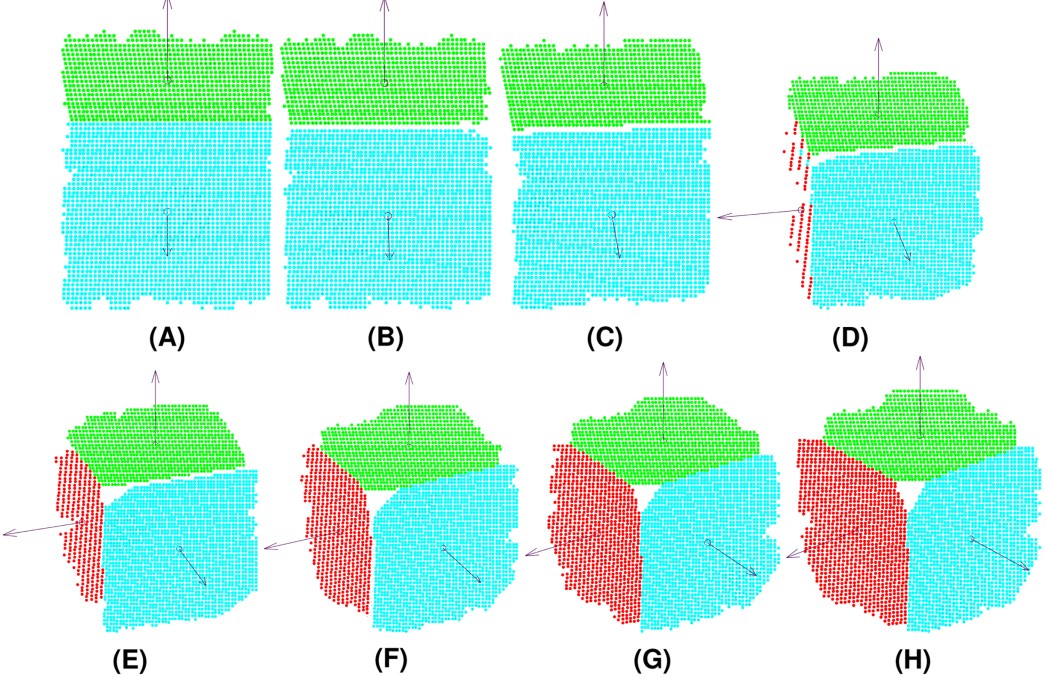

**Figure 5 Point clouds captured by the camera from different views of the target cube: View 1 (A), View 2 (B), View 3 (C), View 4 (D), View 5 (E), View 6 (F), View 7 (G), View 8 (H).**

Note that different levels of Gaussian noise (mean $\mu = 0$ and standard deviation $\sigma$ from 1.00E−05 to 1.00E−04) were added to the synthetic data to evaluate the robustness of the methods (see Fig. 6).

Four performance criteria (see Fig. 7) were used to evaluate the methods. The first two aim at evaluating the global accuracy of the learned models with respect to the ground truth. The last two are used to assess the robustness of the methods in the presence of noisy data:

- *Angle error*: the root mean square (RMS) of the angles (in degrees) composed by the calculated fitted plane and the ground truth for each plane of the model.
- *Model error*: the mean of the differences between the angles (in degrees) among the learned planes that compose the model and the corresponding from the ground truth.

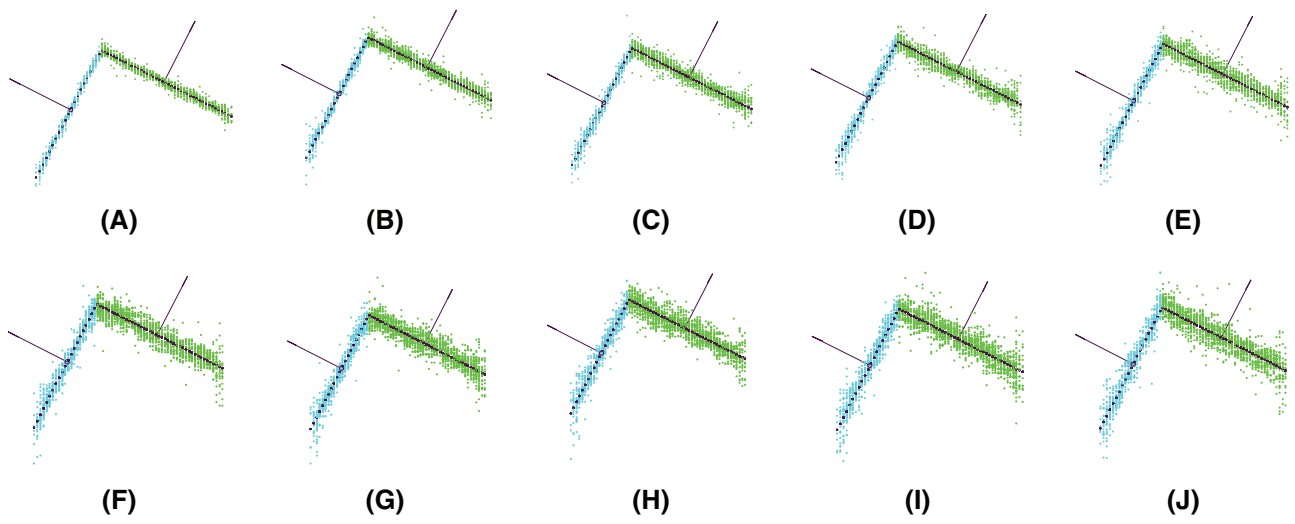

**Figure 6** Synthetic data for different levels of Gaussian noise: $\sigma = 1 \cdot 10^{-5}$ (A), $\sigma = 2 \cdot 10^{-5}$ (B), $\sigma = 3 \cdot 10^{-5}$ (C), $\sigma = 4 \cdot 10^{-5}$ (D), $\sigma = 5 \cdot 10^{-5}$ (E), $\sigma = 6 \cdot 10^{-5}$ (F), $\sigma = 7 \cdot 10^{-5}$ (G), $\sigma = 8 \cdot 10^{-5}$ (H), $\sigma = 9 \cdot 10^{-5}$ (I), $\sigma = 10 \cdot 10^{-5}$ (J).

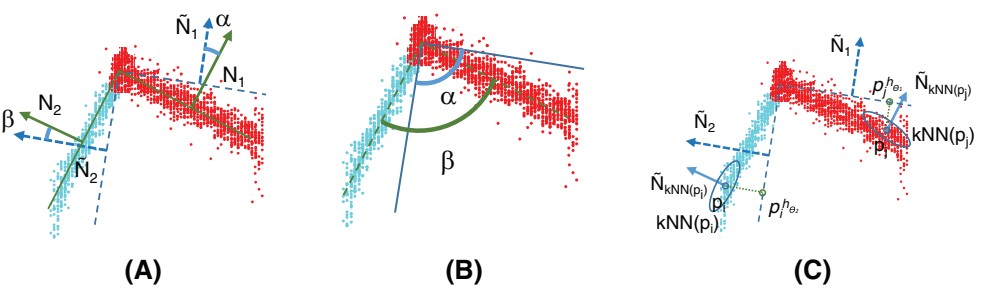

**Figure 7** **Performance criteria used to evaluate methods.** The *angle error* (A) is calculated using the angles between the ground truth (solid line) and the estimated plane (dotted line). In this example, it is $\sqrt{(\alpha^2 + \beta^2)/2}$. The *model error* (B) considers the angles among the planes that compose the object compared with those of the ground truth (solid line). In this example, for an object composed of two planes, it is $\|\alpha - \beta\|/1$. Finally, the *distance error* (C) can be calculated as

$$\sqrt{\left(\left((p_j, \hat{N}_{kNN(p_j)}) - (p_j^{h_{\theta_1}}, \hat{N}_1)\right)^2 + \left((p_i, \hat{N}_{kNN(p_i)}) - (p_i^{h_{\theta_2}}, \hat{N}_2)\right)^2\right) \Big/ 2}.$$

- *Distance error*: the RMS of the Euclidean distance in $\mathbb{R}^6$ between the points and their closest plane. This quantity provides an insight into the residuals of the linear regressions.
- *Cluster error*: the percentage of points incorrectly clustered compared to the ground truth.

## Experimental comparison

MC-LSE is compared to seven other methods as mentioned previously. To assess the impact of the orthogonal constraints, we employed an ordinary LSE and RANSAC

(*Fischler & Bolles, 1981*) as baseline methods. We also compared MC-LSE with state-of-the-art methods such as multi-constraint RANSAC (MC-RANSAC) (*Saval-Calvo et al., 2015a*), RSPD (*Araújo & Oliveira, 2020*), Prior-MLESAC (*Zhao et al., 2020*), Prog-X (*Barath et al., 2020*), and Global-L0 (*Lin et al., 2020*), providing an extensive comparison of our method.

For the sake of comparison, the methods that allow a previous clustering (LSE, MC-RANSAC and RANSAC) use the same pre-clusters as proposed in this paper. Hence, the RANSAC method used in the comparison could be considered as a variant of the related methods based on CC-RANSAC (*Gallo, Manduchi & Rafii, 2011*; *Zhou et al., 2011*; *Zhou et al., 2013*). Because these methods consider the consensus set, the largest connected component of inliers within the spatial neighborhood and within normal vector information coherence, the use of these clusters allows them to be pre-calculated. In other words, the RANSAC checking step to conform to the consensus set considers only the patches previously calculated by k-means that contain coherence in the spatial neighborhood as well as in the normal vector. The RSPD, Prior-MLESAC, Prog-X and Global-L0 methods do not allow the use of the initial pre-clustering because they perform a different pre-treatment that is part of the proposed method such as Prior-MLESAC (calculation of curvature characteristics, normals, etc.), a progressive sampling schema (Prog-X), global optimization approach considering the whole data (Global-L0), or a growth scheme (RSPD).

The results are reported in Table 2, based on which, we can make the following remarks. MC-LSE outperforms all the other methods in terms of the *angle error* and *model error* (being the most reliable performance parameters in terms of model fitting). It is worth noting that this is true regardless of the level of noise added to the data, and the advantage of MC-LSE is even higher with an increasing level of noise. Moreover, our constrained-optimization problem allows us to automatically satisfy the orthogonality requirements (*i.e.* the *model error* is always equal to 0), whereas the other methods suffer from an increasing inability to fulfill the 90-degrees constraint. Specifically, the RSPD method suffers more due to its use of the bottom–up patch growth scheme that detects several planes in the same cluster with a large angle deviation. The two other criteria (*distance error* and *cluster error*) provide some insight into the capacity of the methods to fit models from relevant clusters, to reduce the residuals and to segment the input data. For the former, the two best performing methods are MC-RANSAC and MC-LSE, with very similar results for all noise levels, being the best on average. Except for LSE and RANSAC, all obtained results are very similar because the methods make use, among others, of the point distances to the estimated plane in their optimization processes to perform the fitting. In terms of the *cluster error*, the worst results are obtained by RSPD because it detects several planar patches per plane. LSE and RANSAC obtain the second-worst results because they only make use of partial data (pre-clusters) to obtain the planes and, consequently, have local minima. The methods that consider global constraints (MC-RANSAC, Global-LO, Prog-X and ours) obtain very similar results, with a variation of approximately 1% of errors in the obtained clusters. Finally, it is worth noting that the Prior-MLESAC (also considering the whole data) does not perform a complete assignment

**Table 2 Results (mean ± standard deviation) of the comparative study in terms of four evaluation criteria (see "Experimental Setup") and with respect to various levels of Gaussian noise (from 1.E −05 to 1.E−04).** The results in bold font indicate the best method.

| Noise | Method | Distance error | Cluster error | Angle error | Model error |
|---|---|---|---|---|---|
| 1,00E−05 | LSE | 2.87 ± 0.71 | 6.56 ± 4.22 | 6.18 ± 3.26 | 7.68 ± 3.32 |
| | RANSAC | 1.60 ± 0.15 | 3.88 ± 2.04 | 0.50 ± 0.51 | 0.25 ± 0.17 |
| | MC-RANSAC | 1.60 ± 0.14 | 3.88 ± 2.07 | 0.50 ± 0.34 | 0.35 ± 0.23 |
| | Global-L0 | 2.34 ± 0.15 | 4.87 ± 2.17 | 5.73 ± 1.26 | 5.25 ± 0.89 |
| | RSPD | 9.98 ± 10.01 | 37.38 ± 16.90 | 19.99 ± 18.76 | 21.38 ± 24.87 |
| | Prior-MLESAC | 4.31 ± 0.14 | **1.57 ± 1.20** | 2.49 ± 0.81 | 1.13 ± 0.60 |
| | Prog-X | 3.83 ± 3.30 | 5.37 ± 2.60 | 0.66 ± 0.21 | 0.24 ± 0.45 |
| | MC-LSE (ours) | **1.59 ± 0.15** | 3.87 ± 2.07 | **0.28 ± 0.21** | **0.00 ± 0.00** |
| 2,00E−05 | LSE | 6.88 ± 0.99 | 14.93 ± 4.75 | 20.20 ± 5.76 | 27.68 ± 6.43 |
| | RANSAC | 2.20 ± 0.24 | 4.19 ± 2.05 | 0.86 ± 0.54 | 0.73 ± 0.52 |
| | MC-RANSAC | 2.20 ± 0.18 | 4.36 ± 2.19 | 1.14 ± 0.89 | 0.44 ± 0.14 |
| | Global-L0 | 3.10 ± 0.13 | 4.80 ± 2.15 | 6.97 ± 1.04 | 6.74 ± 1.29 |
| | RSPD | 6.97 ± 6.36 | 39.96 ± 10.94 | 14.42 ± 14.84 | 13.59 ± 17.00 |
| | Prior-MLESAC | 4.47 ± 0.08 | **2.39 ± 2.63** | 2.80 ± 1.00 | 1.38 ± 0.98 |
| | Prog-X | 4.72 ± 4.49 | 5.63 ± 2.31 | 0.58 ± 0.12 | 0.20 ± 0.35 |
| | MC-LSE (ours) | **2.17 ± 0.21** | 4.18 ± 2.05 | **0.32 ± 0.13** | **0.00 ± 0.00** |
| 3,00E−05 | LSE | 9.11 ± 0.69 | 20.14 ± 6.46 | 27.81 ± 3.54 | 40.56 ± 5.81 |
| | RANSAC | 3.00 ± 0.82 | 5.42 ± 1.66 | 1.18 ± 0.57 | 1.27 ± 0.79 |
| | MC-RANSAC | 2.85 ± 0.30 | 5.14 ± 1.92 | 1.33 ± 0.77 | 0.58 ± 0.34 |
| | Global-L0 | 3.82 ± 0.47 | 4.88 ± 2.21 | 8.13 ± 1.69 | 7.85 ± 1.35 |
| | RSPD | 7.99 ± 6.14 | 38.29 ± 9.28 | 19.58 ± 16.53 | 17.40 ± 16.54 |
| | Prior-MLESAC | 4.86 ± 0.46 | **1.70 ± 1.06** | 5.05 ± 3.58 | 2.71 ± 1.56 |
| | Prog-X | 4.56 ± 2.96 | 5.85 ± 2.22 | 0.79 ± 0.20 | 0.20 ± 0.32 |
| | MC-LSE (ours) | **2.69 ± 0.35** | 4.97 ± 1.84 | **0.44 ± 0.23** | **0.00 ± 0.00** |
| 4,00E−05 | LSE | 10.41 ± 0.52 | 23.12 ± 8.90 | 31.42 ± 2.14 | 46.91 ± 2.05 |
| | RANSAC | 7.16 ± 10.64 | 8.94 ± 8.23 | 2.33 ± 1.56 | 11.51 ± 27.25 |
| | MC-RANSAC | 3.15 ± 0.47 | 5.62 ± 2.18 | 1.29 ± 0.70 | 0.61 ± 0.55 |
| | Global-L0 | 4.16 ± 0.29 | 5.41 ± 2.26 | 9.24 ± 0.92 | 8.85 ± 1.97 |
| | RSPD | 5.46 ± 5.01 | 34.65 ± 14.23 | 25.59 ± 22.23 | 27.84 ± 24.16 |
| | Prior-MLESAC | 4.89 ± 0.38 | **1.81 ± 1.00** | 4.16 ± 2.52 | 2.01 ± 1.28 |
| | Prog-X | 6.35 ± 4.70 | 6.05 ± 2.26 | 0.89 ± 0.25 | 0.17 ± 0.36 |
| | MC-LSE (ours) | **3.07 ± 0.40** | 5.46 ± 2.19 | **0.61 ± 0.50** | **0.00 ± 0.00** |
| 5,00E−05 | LSE | 11.22 ± 1.02 | 25.30 ± 10.47 | 33.49 ± 3.75 | 52.54 ± 4.54 |
| | RANSAC | 4.18 ± 1.01 | 6.40 ± 2.56 | 2.87 ± 1.26 | 2.85 ± 2.49 |
| | MC-RANSAC | 3.53 ± 0.35 | 5.68 ± 1.89 | 1.80 ± 0.94 | 0.53 ± 0.31 |
| | Global-L0 | 4.46 ± 0.58 | 5.40 ± 2.42 | 9.90 ± 2.33 | 10.81 ± 4.74 |
| | RSPD | 5.10 ± 1.34 | 36.92 ± 16.45 | 12.35 ± 11.77 | 10.39 ± 10.42 |
| | Prior-MLESAC | 5.24 ± 0.69 | **2.23 ± 0.93** | 5.87 ± 3.75 | 3.29 ± 1.95 |
| | Prog-X | 6.35 ± 4.25 | 6.49 ± 2.30 | 6.86 ± 16.84 | 3.74 ± 9.75 |
| | MC-LSE (ours) | **3.37 ± 0.36** | 5.62 ± 2.27 | **0.57 ± 0.40** | **0.00 ± 0.00** |

| Table 2 (continued) | | | | | |
|---|---|---|---|---|---|
| Noise | Method | Distance error | Cluster error | Angle error | Model error |
| 6,00E−05 | LSE | 12.17 ± 1.15 | 28.06 ± 9.92 | 34.70 ± 3.65 | 58.43 ± 2.95 |
| | RANSAC | 5.26 ± 2.40 | 10.06 ± 10.86 | 4.36 ± 2.64 | 6.72 ± 9.00 |
| | MC-RANSAC | 4.19 ± 0.94 | 6.77 ± 2.33 | 1.86 ± 1.39 | 0.48 ± 0.27 |
| | Global-L0 | 4.82 ± 0.33 | 5.50 ± 2.14 | 10.87 ± 2.65 | 10.08 ± 1.94 |
| | RSPD | 6.11 ± 2.19 | 40.92 ± 10.48 | 28.24 ± 13.33 | 24.01 ± 13.47 |
| | Prior-MLESAC | 5.86 ± 0.86 | **1.94 ± 1.11** | 9.11 ± 3.85 | 5.38 ± 2.72 |
| | Prog-X | 6.45 ± 5.73 | 6.59 ± 2.36 | 1.04 ± 0.53 | 0.13 ± 0.69 |
| | MC-LSE (ours) | **3.65 ± 0.34** | 5.70 ± 2.15 | **0.79 ± 0.47** | **0.00 ± 0.00** |
| 7,00E−05 | LSE | 12.31 ± 1.19 | 29.41 ± 10.83 | 34.33 ± 3.49 | 57.66 ± 3.69 |
| | RANSAC | 18.60 ± 14.16 | 19.09 ± 13.27 | 3.58 ± 1.11 | 41.53 ± 41.22 |
| | MC-RANSAC | 4.19 ± 0.94 | 6.77 ± 2.33 | 1.86 ± 1.39 | 0.48 ± 0.27 |
| | Global-L0 | 5.09 ± 0.58 | 5.63 ± 2.09 | 11.08 ± 1.62 | 10.46 ± 2.41 |
| | RSPD | 4.86 ± 0.89 | 38.50 ± 10.47 | 28.86 ± 18.97 | 24.11 ± 16.26 |
| | Prior-MLESAC | 6.17 ± 1.10 | **2.17 ± 0.96** | 10.45 ± 6.38 | 6.48 ± 6.14 |
| | Prog-X | 6.06 ± 4.01 | 6.88 ± 2.09 | 5.78 ± 12.65 | 1.46 ± 5.35 |
| | MC-LSE (ours) | **3.90 ± 0.44** | 6.01 ± 2.16 | **0.69 ± 0.31** | **0.00 ± 0.00** |
| 8,00E−05 | LSE | 13.10 ± 1.37 | 30.40 ± 11.50 | 35.22 ± 3.61 | 61.73 ± 4.16 |
| | RANSAC | 12.27 ± 13.33 | 14.42 ± 14.18 | 4.51 ± 1.81 | 24.33 ± 36.41 |
| | MC-RANSAC | **4.04 ± 0.50** | 5.95 ± 2.03 | 2.54 ± 2.49 | 0.58 ± 0.39 |
| | Global-L0 | 5.34 ± 0.61 | 6.18 ± 2.29 | 21.11 ± 26.03 | 12.18 ± 4.82 |
| | RSPD | 5.39 ± 1.50 | 41.94 ± 9.84 | 34.65 ± 19.54 | 27.89 ± 12.33 |
| | Prior-MLESAC | 5.61 ± 0.37 | **3.17 ± 1.38** | 6.96 ± 1.72 | 4.49 ± 2.26 |
| | Prog-X | 4.80 ± 1.07 | 8.64 ± 2.84 | 1.50 ± 0.71 | 0.17 ± 0.54 |
| | MC-LSE (ours) | 4.18 ± 0.43 | 6.56 ± 2.81 | **0.67 ± 0.46** | **0.00 ± 0.00** |
| 9,00E−05 | LSE | 14.18 ± 1.75 | 31.80 ± 12.02 | 35.78 ± 4.42 | 64.16 ± 3.82 |
| | RANSAC | 10.99 ± 11.82 | 15.46 ± 15.30 | 8.53 ± 5.03 | 18.43 ± 29.55 |
| | MC-RANSAC | 4.54 ± 0.99 | 6.89 ± 2.76 | 3.00 ± 2.59 | 0.63 ± 0.48 |
| | Global-L0 | 5.71 ± 0.33 | 6.13 ± 2.27 | 21.49 ± 27.30 | 10.65 ± 2.35 |
| | RSPD | 6.82 ± 3.74 | 42.32 ± 9.64 | 37.27 ± 17.87 | 29.12 ± 7.44 |
| | Prior-MLESAC | 6.61 ± 1.62 | **3.76 ± 3.06** | 22.89 ± 39.52 | 8.78 ± 9.16 |
| | Prog-X | 5.18 ± 0.96 | 7.88 ± 2.72 | 10.31 ± 16.30 | 4.81 ± 10.03 |
| | MC-LSE (ours) | **4.41 ± 0.45** | 7.16 ± 2.82 | **0.71 ± 0.26** | **0.00 ± 0.00** |
| 1,00E−04 | LSE | 14.93 ± 1.83 | 34.34 ± 11.28 | 36.24 ± 5.04 | 66.21 ± 3.72 |
| | RANSAC | 8.65 ± 5.68 | 12.58 ± 12.01 | 5.25 ± 2.48 | 7.91 ± 9.46 |
| | MC-RANSAC | **4.49 ± 0.51** | 6.97 ± 2.37 | 2.48 ± 1.36 | 1.02 ± 0.35 |
| | Global-L0 | 5.76 ± 0.55 | 6.81 ± 2.46 | 12.29 ± 3.86 | 10.73 ± 2.51 |
| | RSPD | 5.45 ± 1.17 | 41.85 ± 9.84 | 45.35 ± 23.14 | 24.45 ± 12.14 |
| | Prior-MLESAC | 5.93 ± 0.53 | **3.16 ± 1.59** | 8.21 ± 2.34 | 5.32 ± 4.52 |
| | Prog-X | 5.50 ± 1.30 | 8.42 ± 3.12 | 6.11 ± 11.58 | 4.08 ± 9.80 |
| | MC-LSE (ours) | 5.05 ± 1.49 | 8.00 ± 2.66 | **1.54 ± 1.71** | **0.00 ± 0.00** |

(Continued)

| Noise | Method | Distance error | Cluster error | Angle error | Model error |
|-------|--------|----------------|---------------|-------------|-------------|
| Average | LSE | 10.72 ± 1.12 | 24.41 ± 9.04 | 29.54 ± 3.87 | 48.36 ± 4.05 |
| | RANSAC | 7.39 ± 6.02 | 10.04 ± 8.22 | 3.40 ± 1.75 | 11.55 ± 15.69 |
| | MC-RANSAC | 3.48 ± 0.53 | 5.80 ± 2.21 | 1.78 ± 1.29 | 0.57 ± 0.33 |
| | Global-L0 | 4.46 ± 0.40 | 5.56 ± 2.25 | 11.68 ± 6.87 | 9.36 ± 2.43 |
| | RSPD | 6.41 ± 3.84 | 39.27 ± 11.81 | 26.63 ± 17.70 | 22.02 ± 15.46 |
| | Prior-MLESAC | 5.39 ± 0.62 | **2.39 ± 1.49** | 7.80 ± 6.55 | 4.10 ± 3.12 |
| | Prog-X | 5.38 ± 3.28 | 6.78 ± 2.48 | 3.45 ± 5.94 | 1.52 ± 3.76 |
| | MC-LSE (ours) | **3.41 ± 0.46** | 5.75 ± 2.30 | **0.66 ± 0.47** | **0.00 ± 0.00** |

of the points of the scene to a cluster, and the results are better than the others. Thus, although MC-LSE does not obtain the best results in terms of clusters, it is capable of achieving the best fit.

Our objective is to compare MC-LSE with other methods, in terms of not only its error, but also its computational cost. It is worth remembering that MC-LSE is based on an iterative process that is repeated until convergence. Formally, convergence is reached if no reassignment is performed between the two iterations. However, note that if only a few points are assigned to the wrong cluster, MC-LSE is not prevented from learning a good model. To evaluate this behavior, we performed three additional experiments consisting of stopping MC-LSE when no more than 3%, 5% and 10% of the points change between two iterations. Figure 8 shows a plot of the joint behavior of MC-LSE in terms of both the running time (in seconds) and angle error in the four settings: convergence until no (red ball), no more than 3% (cyan ball), no more than 5% (brown ball) and no more than 10% (blue point) of reassignment. We also report the results of MC-RANSAC (blue diamond), RANSAC variant (purple square), Global-L0 (orange right arrow), RSPD (purple left arrow), Prior-MLESAC (green triangle) and Prog-X (blue star). It is important to note that LSE, RANSAC, Prior-MLESAC, MC-RANSC and MC-LSE were implemented in MATLAB 2019, whereas Global-L0, RSPD and Prog-X were implemented in C++, as described by the authors of the papers. Therefore, the latter obtains results in a faster time than the MATLAB version. We can see that by relaxing the convergence constraint, MC-LSE is still very accurate with an execution time very close to the minimum provided by Global-L0, Prog-X and RANSAC (not counting LSE and RSPD as the fastest but worst results). The RANSAC variant is faster but much less accurate, whereas MC-RANSAC yields a small error, but is much more time-consuming. The implementations of Prog-X and Global-L0 (in C++) provide similar processing times with better performance for Prog-X. Although the median of Prog-X is similar to that of MC-LSE, the results are very scattered, ranging from less than 1 degree to more than 10 for certain acquisitions.

## Specific analysis of MC-LSE

Once the comparative analysis is performed, in this section, we focus on the analysis of MC-LSE. First, the pre-clustering results obtained by the clustering-based unsupervised

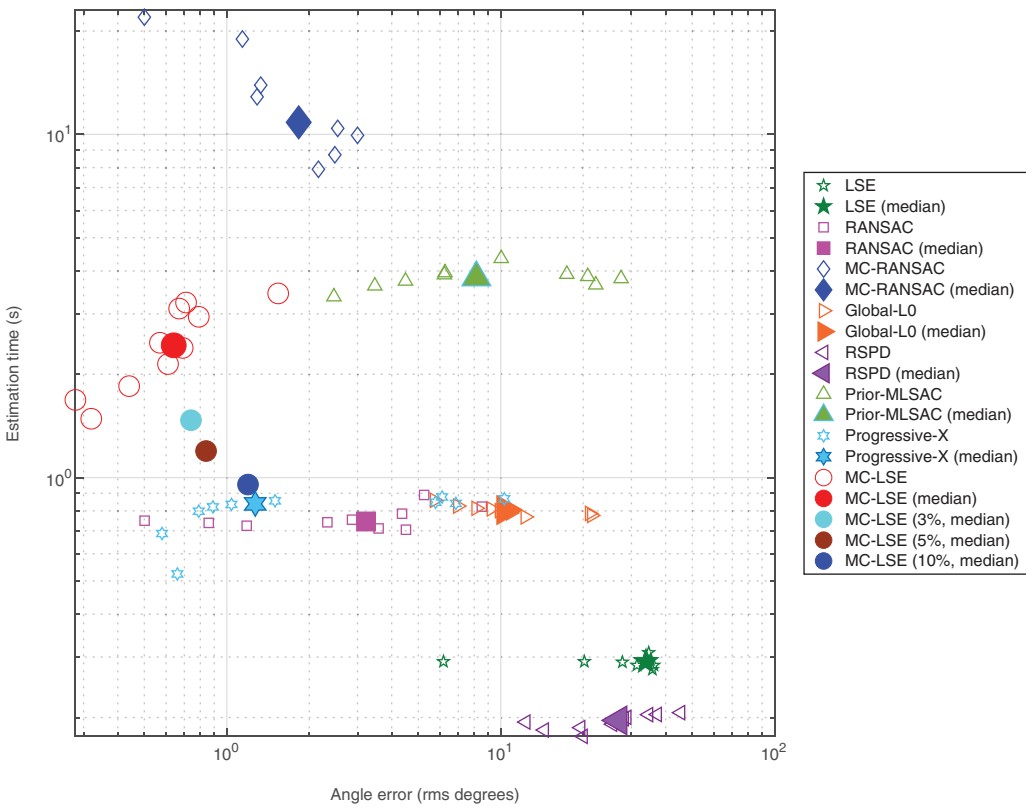

**Figure 8 Comparison of plane estimation methods with respect to processing time and angle error.**

step (see "Clustering Step") are analyzed. Subsequently, the behavior of MC-LSE according to the number of iterations and processing time is analyzed. Next, view V5 is analyzed in detail because the results are the most different compared to the rest of the views. The main difference from the other views is that the data are highly imbalanced.

### Pre-clustering results

The first step of MC-LSE is to obtain a set of pre-clusters in the scene. The calculation of this first step is critical in the final result because the entire process starts from it. Therefore, in this section, we analyze the results obtained for the four most representative clustering methods: Gaussian mixture model-based expectation and maximization (GMM-EM) (*Ari & Aksoy, 2010*), hierarchical clustering (HC) (*Cai et al., 2014*), self-organizing maps (SOM) (*Mingoti & Lima, 2006*), and k-means (*Forgy, 1965*). It should be noted that the clustering performed with SOM applies k-means to the map generated from the set $\tilde{P}$ of normal vectors and 3D points (see "Clustering Step").

Table 3 lists the results of the entire MC-LSE for the evaluation criteria (see "Experimental Setup") with respect to various levels of Gaussian noise (from 1.E−05 to 1.E−04) for the eight tested different views, except for the *model error*. It is not calculated as the proposed method meets the geometrical constraints (see results in Table 2). Additionally, the *cluster error* is calculated for the pre-clusters resulted of the unsupervised methods (*pre-cluster error* in the Table 3).

**Table 3 Results (mean ± standard deviation) of the pre-clustering methods study over four evaluation criteria and with respect to various levels of Gaussian noise (from 1.E−05 to 1.E−04).** The results in bold indicate the best method.

| Noise | Clustering | Pre-cluster error | Cluster error | Distance error | Angles error |
|---|---|---|---|---|---|
| 1.00E−05 | GMM-EM | 5.34 ± 7.10 | 5.69 ± 8.28 | 2.99 ± 4.23 | 5.22 ± 13.44 |
| | HC | 4.57 ± 2.12 | 4.08 ± 1.81 | 1.71 ± 0.35 | 0.57 ± 0.50 |
| | SOM | 4.27 ± 2.28 | 3.86 ± 2.11 | 1.70 ± 0.34 | 0.57 ± 0.49 |
| | K-means | **4.15 ± 2.20** | **3.87 ± 2.07** | **1.59 ± 0.15** | **0.28 ± 0.21** |
| 2.00E−05 | GMM-EM | **4.03 ± 2.50** | **3.99 ± 2.19** | **2.05 ± 0.10** | **0.53 ± 0.23** |
| | HC | 6.85 ± 3.30 | 4.62 ± 1.65 | 2.25 ± 0.35 | 0.68 ± 0.44 |
| | SOM | 6.65 ± 2.96 | 4.24 ± 2.05 | 2.16 ± 0.20 | 0.55 ± 0.26 |
| | k-means | 6.32 ± 2.55 | 4.18 ± 2.05 | 2.17 ± 0.21 | 0.32 ± 0.13 |
| 3.00E−05 | GMM-EM | **6.92 ± 8.14** | 8.37 ± 11.37 | 3.84 ± 3.92 | 4.95 ± 10.98 |
| | HC | 9.38 ± 3.73 | 5.48 ± 2.78 | 3.29 ± 1.73 | 1.90 ± 2.72 |
| | SOM | 10.85 ± 2.54 | 5.17 ± 1.68 | **2.68 ± 0.34** | 0.94 ± 0.47 |
| | k-means | 10.61 ± 1.83 | **4.97 ± 1.84** | 2.69 ± 0.35 | **0.44 ± 0.23** |
| 4.00E−05 | GMM-EM | **5.38 ± 2.98** | **5.25 ± 2.58** | **2.70 ± 0.15** | **0.77 ± 0.65** |
| | HC | 14.48 ± 5.01 | 5.69 ± 2.10 | 3.09 ± 0.40 | 1.08 ± 0.65 |
| | SOM | 14.17 ± 3.36 | 5.68 ± 2.39 | 3.09 ± 0.39 | 1.32 ± 1.14 |
| | k-means | 13.87 ± 2.68 | 5.46 ± 2.19 | 3.07 ± 0.40 | 0.61 ± 0.50 |
| 5.00E−05 | GMM-EM | **9.95 ± 9.49** | 10.47 ± 12.74 | 4.85 ± 3.76 | 8.48 ± 17.08 |
| | HC | 15.34 ± 6.49 | 6.57 ± 4.08 | 4.06 ± 1.95 | 6.89 ± 17.62 |
| | SOM | 15.89 ± 2.42 | **5.55 ± 1.77** | 3.42 ± 0.39 | **0.69 ± 0.26** |
| | k-means | 16.96 ± 3.84 | 5.62 ± 2.27 | **3.37 ± 0.36** | 0.57 ± 0.40 |
| 6.00E−05 | GMM-EM | **7.94 ± 5.95** | 8.36 ± 8.88 | 4.38 ± 2.95 | 6.32 ± 15.05 |
| | HC | 17.74 ± 4.29 | 7.49 ± 4.95 | 4.70 ± 2.47 | 4.29 ± 7.25 |
| | SOM | 20.47 ± 3.72 | 6.81 ± 2.53 | 4.21 ± 1.51 | 2.62 ± 2.77 |
| | K-means | 20.54 ± 3.41 | **5.70 ± 2.15** | **3.65 ± 0.34** | **0.79 ± 0.47** |
| 7.00E−05 | GMM-EM | **9.73 ± 6.61** | 6.64 ± 3.15 | 4.16 ± 1.26 | 5.30 ± 10.97 |
| | HC | 18.79 ± 5.30 | 8.24 ± 6.98 | 4.87 ± 2.66 | 4.20 ± 8.46 |
| | SOM | 21.21 ± 3.91 | 6.51 ± 2.61 | 3.93 ± 0.42 | 1.77 ± 2.11 |
| | K-means | 21.06 ± 3.43 | **6.01 ± 2.16** | **3.90 ± 0.44** | **0.69 ± 0.31** |
| 8.00E−05 | GMM-EM | **7.30 ± 4.39** | **6.91 ± 3.13** | 4.44 ± 1.16 | 2.70 ± 3.35 |
| | HC | 24.56 ± 7.63 | 12.11 ± 13.35 | 6.09 ± 3.76 | 7.66 ± 15.04 |
| | SOM | 24.45 ± 3.16 | 8.28 ± 3.39 | 4.75 ± 1.38 | 3.13 ± 3.99 |
| | K-means | 24.40 ± 3.39 | 6.56 ± 2.81 | **4.18 ± 0.43** | **0.67 ± 0.46** |
| 9.00E−05 | GMM-EM | **11.15 ± 7.80** | 9.25 ± 5.17 | 5.66 ± 2.53 | 7.01 ± 10.43 |
| | HC | 27.42 ± 4.47 | 7.80 ± 2.80 | 4.44 ± 0.46 | 1.92 ± 1.41 |
| | SOM | 25.63 ± 4.54 | 8.50 ± 3.39 | 4.98 ± 1.50 | 2.96 ± 2.92 |
| | K-means | 25.54 ± 3.99 | **7.16 ± 2.82** | **4.41 ± 0.45** | **0.71 ± 0.26** |
| 1.00E−04 | GMM-EM | **6.75 ± 4.54** | **7.86 ± 4.63** | 4.76 ± 1.80 | 4.60 ± 5.93 |
| | HC | 29.44 ± 5.94 | 8.16 ± 2.70 | 5.39 ± 1.65 | **2.49 ± 2.40** |
| | SOM | 26.19 ± 5.63 | 8.55 ± 3.03 | 5.07 ± 1.43 | 2.86 ± 3.08 |
| | k-means | 27.01 ± 5.04 | 8.00 ± 2.66 | **5.05 ± 1.49** | 1.54 ± 1.71 |
| Average | GMM-EM | **7.53 ± 6.11** | 7.21 ± 6.39 | 3.90 ± 2.23 | 4.59 ± 9.13 |
| | HC | 15.46 ± 4.70 | 6.90 ± 4.50 | 3.83 ± 1.57 | 3.24 ± 6.01 |
| | SOM | 15.96 ± 3.21 | 6.07 ± 2.44 | 3.44 ± 0.72 | 1.61 ± 1.60 |
| | K-means | 15.94 ± 3.04 | **5.75 ± 2.30** | **3.40 ± 0.46** | **0.66 ± 0.47** |

**Table 4 Results (mean ± standard deviation) of the pre-clustering method considering all experimental data (for different various levels of Gaussian noise and viewpoints).** The results in bold font indicate the best method.

|  | Normals | Normals & points | Points |
|---|---|---|---|
| Pre-cluster error (percentage) | 16.416 ± 2.880 | **14.189 ± 3.104** | 17.365 ± 12.024 |
| Cluster error (percentage) | **5.693 ± 2.279** | 5.753 ± 2.248 | 17.015 ± 13.151 |
| Distance error (rms distance) | 3.246 ± 0.351 | **3.236 ± 0.353** | 7.635 ± 3.183 |
| Angle error (rms degrees) | **0.640 ± 0.246** | 0.677 ± 0.536 | 14.639 ± 13.867 |

First, the error of the clusters generated in the first step (*pre-cluster error*) and the result of the whole process (*cluster error*) are analyzed. It can be seen that the best method is GMM-EM, which optimizes the GMM distributions. This is the best result in all cases except for the lowest noise level (1.E−05). Because the noise used for testing is Gaussian noise, it can explain the very good results. However, in comparison to the final result of the process, it can be seen that all methods, on average, generate similar results, with k-means being the best. In addition, the GMM-EM algorithm provides the best results for noise levels of 2.E−05, 4.E−05, 8.E−05 and 1.E−04. As can be seen, the GMM-EM and HC methods show an excessively high deviation, resulting in very good results in some views and very bad results in others. Therefore, good pre-cluster results do not guarantee a good final result because it depends on the set of points in the pre-cluster that may be in different planes of the real model. This case will be analyzed later with View 5 in "Analysis of View V5: A Difficult Case".

Finally, according to the *distance error* variable for all noise levels, similar results were obtained (again, except for GMM-EM with high standard deviations). In any case, the *angle error* and *model error* could be the most reliable variables for defining the accuracy of the method. However, as previously mentioned, because the method meets the geometrical constraints, the *model error* is 0. However, *angle error* shows that both SOM and k-means provide the best results, with k-means being the one with the best average error (0.66 degrees). In all cases for k-means, the RMS error is less than 1 degree except for the maximum level of noise (1.E−04) and having small deviations, indicating its capability to obtain very accurate results regardless of the viewpoint.

To conclude the study of the pre-clustering results, the influence of the input data on the pre-clustering calculation step was also analyzed. As discussed in "Clustering Step", the clustering algorithm considers not only the original features, $p_i = (p_i^x, p_i^y, p_i^z)$, but also the 3D-normal vector, $\tilde{N}_{kNN(p_i)}$, of set $kNN(p_i)$ of the nearest neighbors of $p_i$ to conform to the corresponding points, $\tilde{p}_i \in \mathbb{R}^6$, used as inputs of the clustering algorithm. In the Table 4, the results of using only $\tilde{N}_{kNN(p_i)}$ (Normals), only original features $p_i$ or both, $\tilde{p}_i$ (Normals and points), according to the variables studied previously are shown. As can be seen, the use of normal vectors improves the performance of MC-LSE. In addition, the most reliable parameter *angle error* is less than 1 degree using normal vectors, and more than 14 degrees if only the original features $p_i$ are used. In this case, MC-LSE is not capable of extracting accurate plane models. From these data, it can be concluded that the use

**Table 5 Convergence and processing time are evaluated according to the following features:** *Iterations* **as the number of iterations used to learn the models.** *Fitting time* as the time (in seconds) required to fit the linear models and the *Reassignment time* as the time (in seconds) used to recalculate the clusters at each iteration with respect to the learned planes.

| Noise | Iterations | Fitting time (s) | Reassignment time (s) |
|---|---|---|---|
| 1.00E−05 | 3.3 ± 1.035 | 0.729 ± 0.198 | 0.956 ± 0.663 |
| 2.00E−05 | 3.0 ± 0.535 | 0.665 ± 0.138 | 0.820 ± 0.510 |
| 3.00E−05 | 3.8 ± 0.463 | 0.831 ± 0.081 | 1.016 ± 0.623 |
| 4.00E−05 | 4.3 ± 1.669 | 0.924 ± 0.327 | 1.217 ± 0.935 |
| 5.00E−05 | 5.0 ± 2.619 | 1.102 ± 0.546 | 1.369 ± 1.169 |
| 6.00E−05 | 5.6 ± 2.774 | 1.288 ± 0.599 | 1.653 ± 1.393 |
| 7.00E−05 | 4.9 ± 1.959 | 1.060 ± 0.319 | 1.327 ± 0.968 |
| 8.00E−05 | 6.4 ± 2.134 | 1.436 ± 0.472 | 1.672 ± 1.161 |
| 9.00E−05 | 6.8 ± 1.669 | 1.491 ± 0.284 | 1.745 ± 1.077 |
| 1.00E−04 | 6.8 ± 2.605 | 1.520 ± 0.542 | 1.920 ± 1.506 |

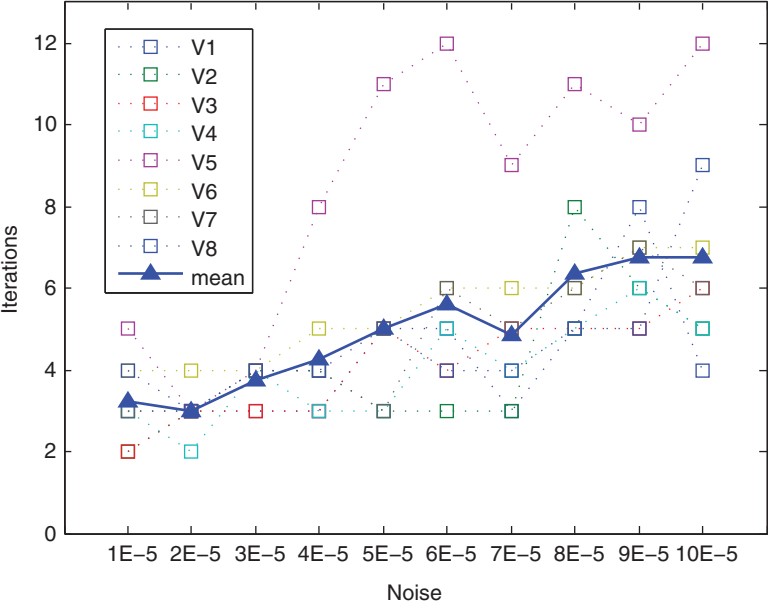

**Figure 9 Iterations needed for method to converge.**

of normal vectors is very important in the clustering step and, consequently, in the results obtained by the method. Furthermore, in this case, although practically the same results are obtained, it is better to use $\tilde{p}_i$ as inputs to separate planes that have the same orientation but are in different positions of the scene. We explain this case in the room reconstruction of the case study described in "Room Reconstruction".

## Convergence and processing time results

The method converges very quickly, as shown in Table 5 (mean and standard deviation) and Fig. 9. The behavior in terms of convergence of the method is similar for the different

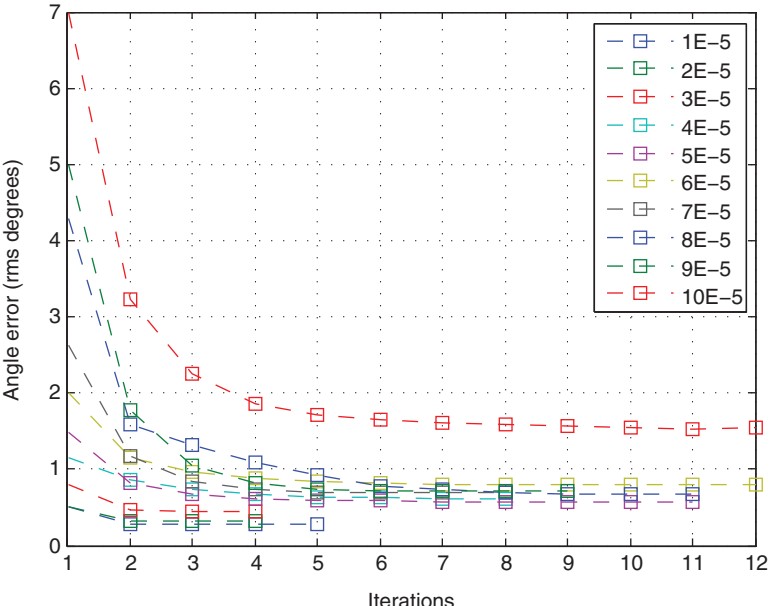

**Figure 10 Angle error for each iteration of the method.**

levels of noise and views, except for view V5. It can converge in six or fewer iterations, even for high levels of noise (up to 7E−5). For the highest levels of noise (from 8E−5 to 1E−4) slightly increased, with a mean of approximately 6.5 iterations for this range. The worst case is view V8 for the 1E−4 level of noise, reaching up to nine iterations. Regarding view V5, it requires the largest number of iterations to converge for all levels of noise (except for level 2E−5), reaching the maximum for noises 6.0E−05 and 10.0E−05, requiring a total of 12 iterations. Hence, this view is analyzed in depth in the following subsection.

Given only the number of iterations to converge, the variation in the quality of the solution in each iteration is not considered. Hence, it is interesting to analyze the number of iterations according to the *angle error* as the most precise accuracy parameter studied in the previous section. Figure 10 shows the behavior of the variable with respect to the number of iterations. It represents the mean of the quality variable for different views at a specific level of noise. The error drops significantly in the first iteration, with a performance similar to an exponential decrease. Moreover, the error does not decrease significantly after approximately 4–5 iterations for any level of noise.

It is important to remember that the method converges when the number of points that are reassigned for a cluster with respect to the previous iteration is lower than a certain threshold. In the experiments, the threshold is 0, which means that the method does not stop as long as changes in the assignment continue to occur. However, as shown in Fig. 11 for a noise level of 10.0E−05 (the level with most iterations), in the first four iterations, the percentage of points that change is less than 1% except for views V5 and V6. In iteration 5, view V6 decreases from 2.62% to 0.75%; and view V5 decreases from 3.58% to 2.08%. From iteration 6 onwards, in all cases, it is less than 1%. After iteration 8, the change in view V5 was 0.3.

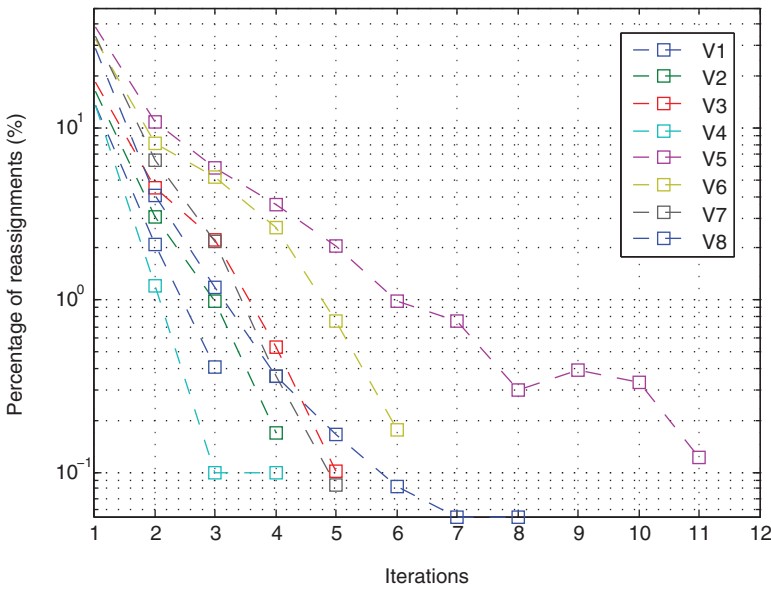

**Figure 11  Iterations.**

The *fitting time* and *reassignment time* are completely dependent on the iterations made by the method, and they are not affected by noise. Specifically, the former in the experiments is also not dependent on the number of points in the input data. The method can provide a model fitted into the input data within 0.2463 ± 0.0151 s per iteration. The latter is dependent on the number of planes in which the model to be fitted is composed of the number of input data.

### Analysis of view V5: a difficult case

The behavior of the results for view V5 is different from that of the rest. The face $P_2$ in view of V5 (Fig. 5E) is highly imbalanced with respect to the other faces $P_1$ and $P_3$. Moreover, the normal vector of $P_2$ is approximately orthogonal to the point of view of the camera. In this section, we focus on the results for the highest level of noise (Fig. 12).

First, the *pre-clusters* are calculated by the clustering-based unsupervised step (see "Clustering Step") could be analyzed in Fig. 12A. Although *pre-clusters* of planes $P_1$ (green) and $P_3$ (blue) are mainly distributed around the corresponding planes, the points pre-clustered as $P_2$ (red) are distributed on planes $P_2$ and $P_3$. These results can be analyzed quantitatively in Table 6 (left). The points predicted as $P_2$ are 33.7% for the actual face, whereas 55% of the predictions are for points belonging to $P_3$. In other words, the pre-cluster is distributed in two faces $P_2$ and $P_3$, with most of the points incorrectly classified.

For the improved initialization of the planes, landmarks are selected using 50% of the points for each cluster $P_i$ closest to their centroid. The results are shown in Fig. 12B and quantitatively analyzed in Table 6 (middle). The landmarks for planes $P_1$ and $P_3$ are well calculated, but the results for $P_2$ are even worse. The points predicted as $P_2$ are 33.3% for the actual face, whereas 63.9% of the predictions are for points belonging to $P_3$.

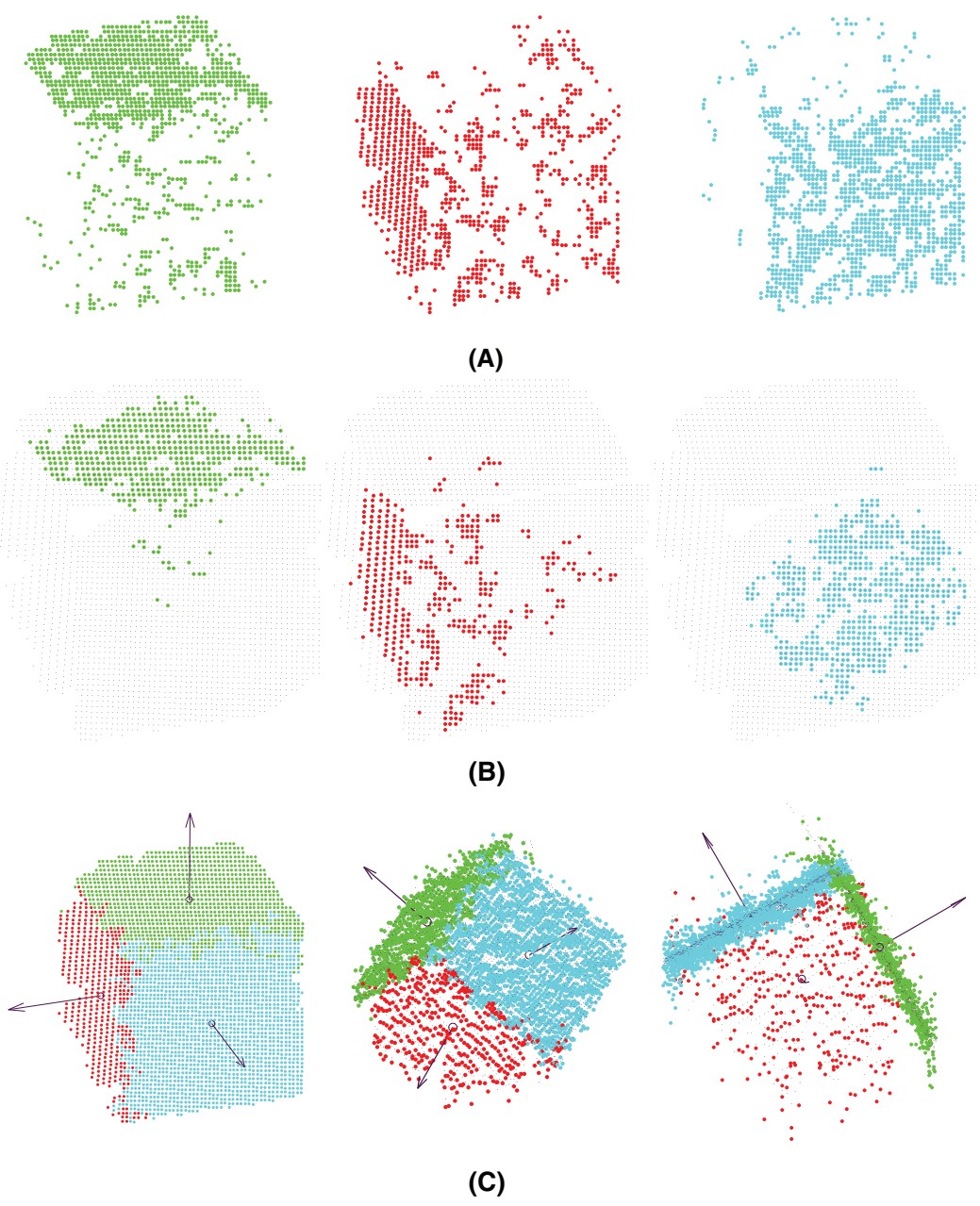

**Figure 12** Noisy points (10E−5) captured in view V5, preclusters (A), landmarks (B) and fitted model (C) for view 5. Planes $P_1$ (green), $P_2$ (red) and $P_3$ (blue).

Finally, Fig. 12C shows the fitted model for the method after convergence and Table 6 (right) the confusion matrix. The model was fitted perfectly to the data. The errors for the clusters were mainly distributed at the intersection of the planes and at the edges of the faces. For example, 21% of the points predicted as $P_2$ for actual $P_3$ points are those at the edges. Because the number of points of the cluster $P_2$ is less than the others, for approximately 10% of the captured points, the relative error is higher.

**Table 6 Confusion matrices for clustering of input data from view five and noise level 10.0E−05.** Pre-clustered based on k-means (left), landmarks (middle) and results obtained by MC-LSE (right).

| | | Predicted | | | | | Predicted | | | | | Predicted | | |
|---|---|---|---|---|---|---|---|---|---|---|---|---|---|---|
| | | PA1 | PA2 | PA3 | | | PA1 | PA2 | PA3 | | | PA1 | PA2 | PA3 |
| **Actual** | PA1 | 69.4% | 11.3% | 8.1% | **Actual** | PA1 | 93.0% | 2.8% | 0.5% | **Actual** | PA1 | 90.7% | 6.8% | 1.3% |
| | PA2 | 0.9% | 33.7% | 0.9% | | PA2 | 0.0% | 33.3% | 0.0% | | PA2 | 3.3% | 72.4% | 0.3% |
| | PA3 | 29.6% | 55.0% | 90.9% | | PA3 | 7.0% | 63.9% | 99.5% | | PA3 | 9.0% | 20.8% | 98.4% |

## Cases of study: reconstructing scenes

This section presents two case studies to qualitatively evaluate the performance of MC-LSE in realistic situations. Specifically, this experiment consists of registering a set of 3D point clouds using a state-of-the-art method, $\mu$-MAR (*Saval-Calvo et al., 2015b*), which uses multiple 3D planar surfaces to find the transformations between them. To obtain an accurate model, the estimation of transformations to align views is critical. $\mu$-MAR uses the model of the planes instead of the actual 3D data to reduce noise effects and, hence, improve registration. By using multiple non-coplanar planes, the method estimates the correspondences between views and calculates the transformation using the normals and centroids of the planes. The method uses the normals of both the fixed and moving sets of plane models to determine the rotation. Next, the translation is iteratively calculated by projecting the centroids of the moving set to the fixed set and minimizing the distances. For more details, refer to the original paper (*Saval-Calvo et al., 2015b*). Because the final registration result highly depends on the accuracy of the planar models, the better those models are, the more accurate the result will be.

"Object Reconstruction" shows the reconstruction of two objects that are part of the original dataset of the $\mu$-MAR. As in the original study, the evaluation was performed by visual inspection because there is no ground truth to compare with. Next, "Room Reconstruction" presents a reconstruction of a scene composed of multiple orthogonal planes (walls and ceiling), the models of which have been estimated using MC-LSE, and the $\mu$-MAR has been used to align the views.

### Object reconstruction

As previously mentioned, here, we show the reconstruction of two objects using a 3D planar marker registration method, where both MC-RANSAC and MC-LSE methods are used to estimate the planes of the markers (cubes) under orthogonal constraints. The planes extracted for the cubes were registered using the $\mu$-MAR method. Finally, the transformations calculated for the cube are applied to the point clouds of the objects to reconstruct them.

The objects were a bomb and a taz toy. Figure 13 shows the two objects and the configuration with the 3D markers around them. The set-up includes a turntable that rotates using a stepper motor and an RGB-D camera that takes color and depth images in every step of the motor. The table was covered with a blue fabric to ease the segmentation of the objects and markers.

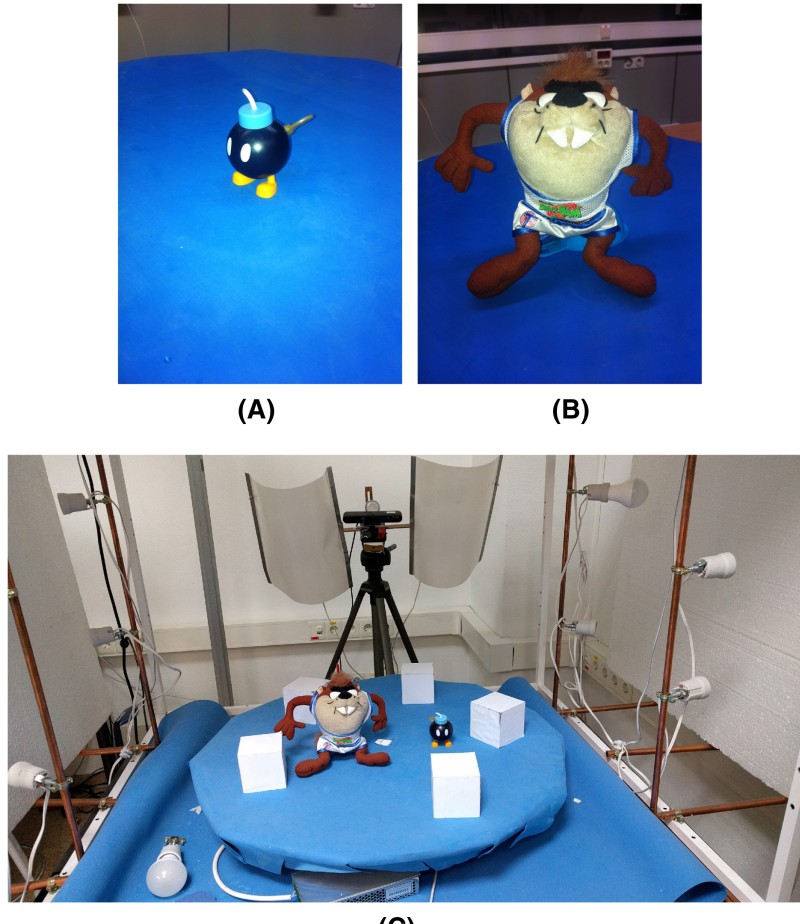

**Figure 13 Objects used for reconstruction.** From left to right, the bomb toy (A), the taz toy (B), and finally the set-up (C) of the toys with the 3D markers.

After applying the $\mu$-MAR registration method, we obtained an aligned point cloud. The result is shown in Fig. 14, where the markers were removed from the scene for a clearer interpretation. Although both methods provide good results, small improvements can be observed using MC-LSE. In the taz toy, the right leg showed better alignment. The bomb toy in Fig. 14B shows a more rounded shape. For better visualization, Fig. 15 shows the details of the reconstruction. The first row presents a zoomed view of the leg in the Taz toy, where the MC-LSE planes achieve better reconstruction than the planes of MC-RANSAC. The second row shows the bomb toy, where the object reconstructed by MC-LSE is more round, and the eyes are more defined (*i.e.* the views are better aligned).

Figure 16 shows a bottom view where the markers (cubes) have not been removed to provide another reference to evaluate the accuracy of the alignment. In general, the markers are better aligned in MC-LSE, with the right cube a clear example with a more compact shape.

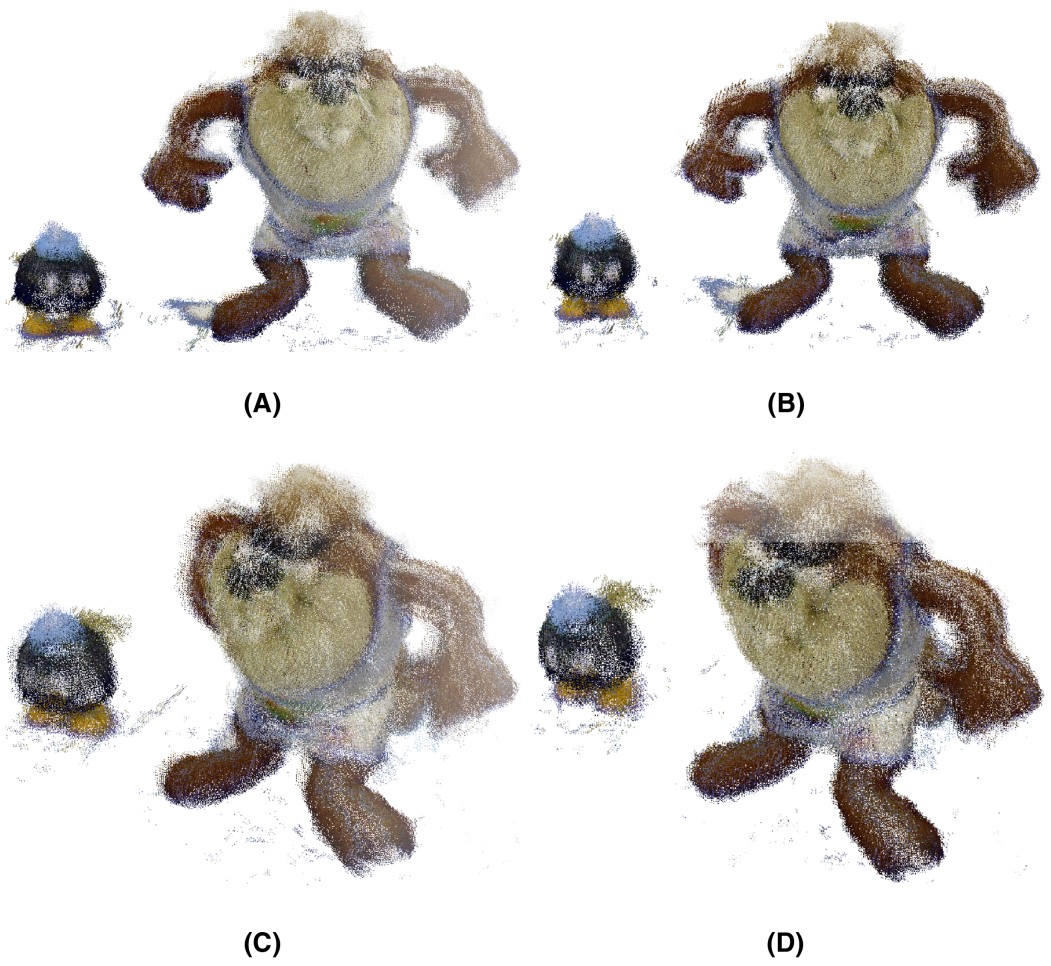

**(A)**     **(B)**

**(C)**     **(D)**

**Figure 14** **Taz and bomb toys reconstructed using MC-RANSAC and MC-LSE.** The top row shows the front view for MC-RANSAC (A) and MC-LSE (B), and the second row shows the side view for MC-RANSCAC (C) and MC-LSE (D). MC-LSE obtains better alignment.

### Room reconstruction

In the second part of the case study, we present an indoor scene reconstruction composed of multiple orthogonal planes (walls and ceiling) registered using the $\mu$-MAR algorithm, to estimate planes extracted by the proposed method. This application is related to the SLAM problem in robot localization and indoor building reconstruction.

The point cloud was captured using factory calibration for the Kinect camera. The reconstruction is more challenging than in the previous case of study because it has to deal with the same problems as before, but with larger errors; the optical aberration is worse than the fact that most points are closer to the border of the image (Fig. 17B) and the planes are further (about 4 meters in some views) increasing the noise in the point cloud (Fig. 17C).

Figure 18 shows the five acquired data viewpoints of the room. $\mu$-MAR uses pairwise alignment; hence, every two consecutive frames have some planes in common. Overlapped on the images are the clusters corresponding to the planes. For this case study, the

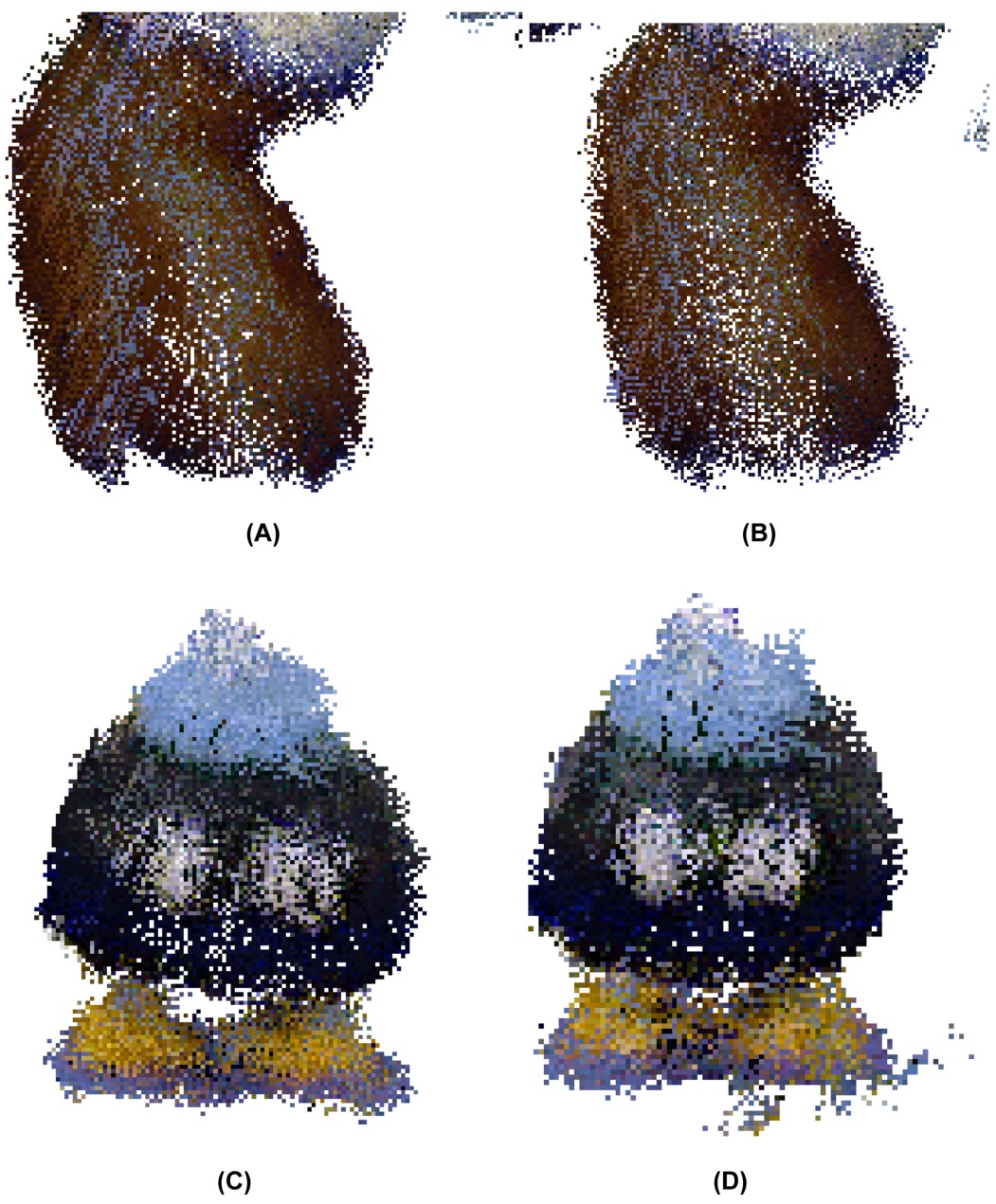

**Figure 15 Detail of the reconstruction for a better evaluation: leg and bomb details with MC-RANSAC (A, C) and MC-LSE (B, D).**

preclusters corresponding to planes have been manually segmented because it is not a core contribution of the study. Moreover, the constraints have been relaxed, not being strictly orthogonal due to the curves present in the planes as shown in Fig. 17B.

The final reconstruction is shown in Fig. 19. As can be seen, the final alignment is correct, and the geometrical constraints are preserved using the plane models estimated by MC-LSE. The capabilities and robustness of the proposed method for estimating planes even in the presence of high noise and optical aberration can be seen in the planes estimated in Fig. 20 from the frame in Fig. 18, which are far from the camera and close to

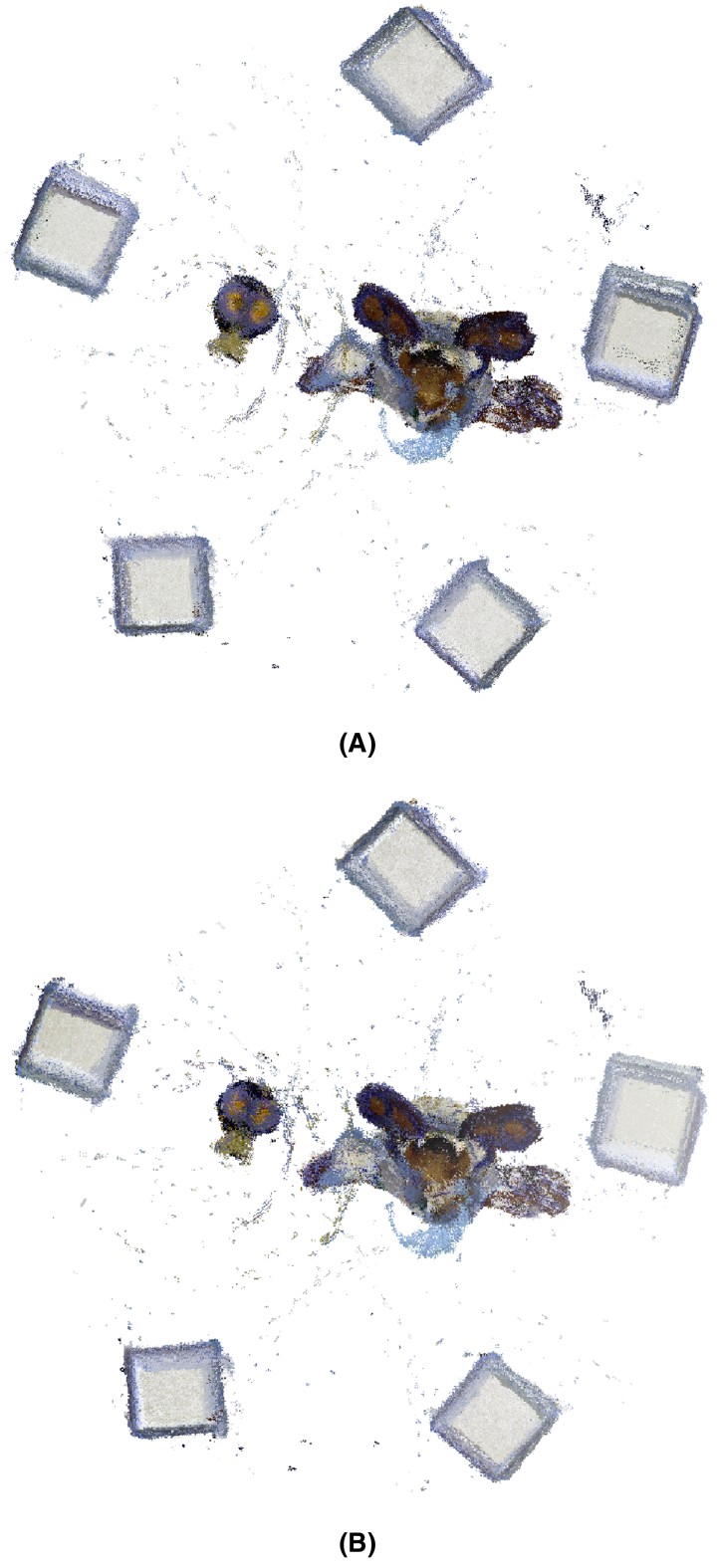

(A)

(B)

**Figure 16 Taz and bomb toys reconstructed with the markers.** The top row shows the bottom view using MC-RANSAC (A), and the second row shows MC-LSE (B), which is the second best aligned.

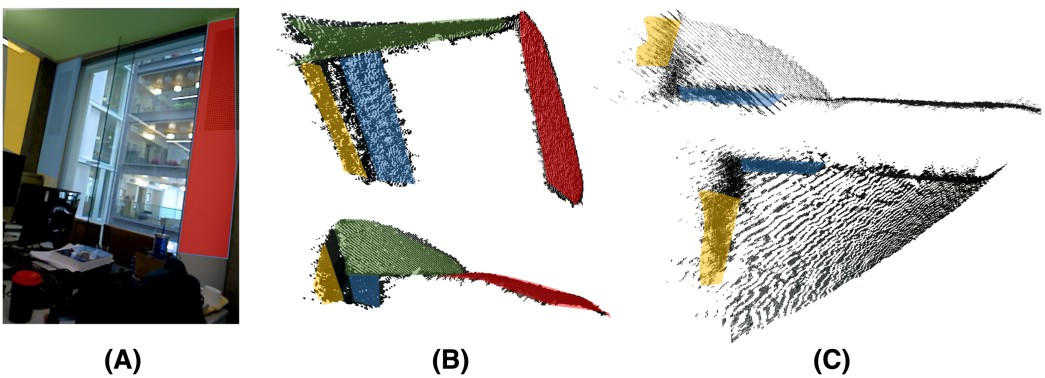

**(A)** **(B)** **(C)**

**Figure 17 Errors in scene acquisition of a room (A).** Optical aberration curves of the green and red planes (B). Distant planes (approximately 4 m), given in orange, are acquired with low accuracy (C), and the noise is more than 20 cm.

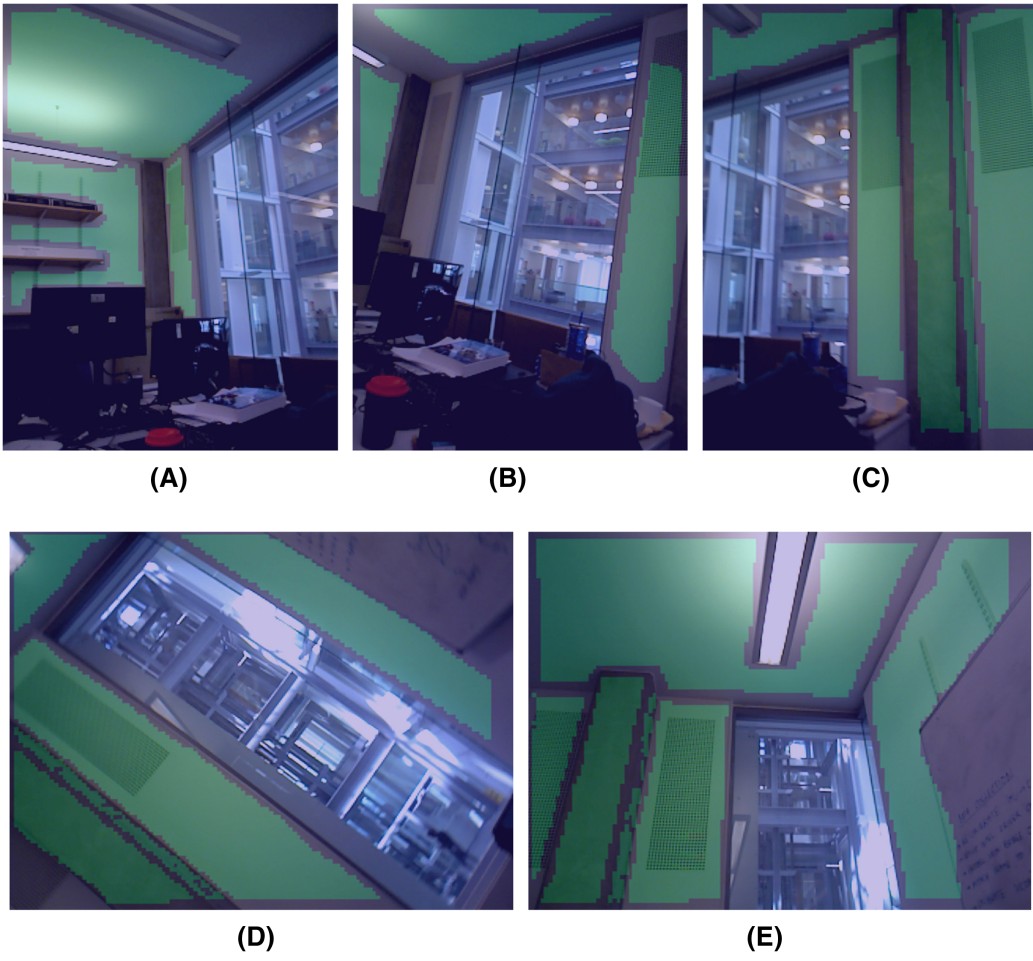

**(A)** **(B)** **(C)**

**(D)** **(E)**

**Figure 18 (A–E) Five frames used for the scene reconstruction experiment.** The color images have segmented planes marked in turquoise.

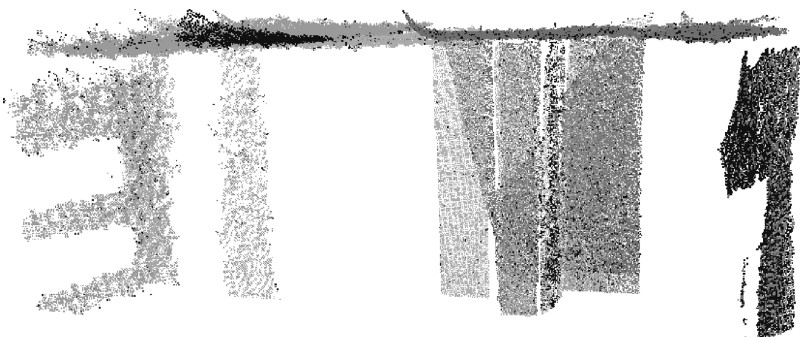

**Figure 19 Full model reconstruction of the scene using the data acquired by the Kinect.**

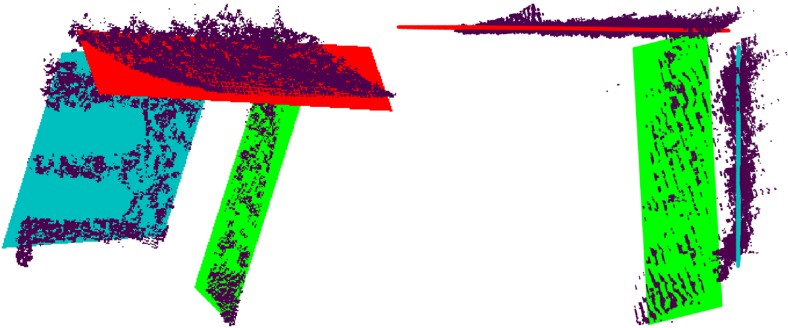

**Figure 20 Detail of a noisy frame (A) in Fig. 18.** The figure shows the three models of the planes in red, turquoise and green properly estimated despite the noise.

the borders in the image; hence, the proposed method is very noisy and curved. In this figure, the three planes of the frame are shown along with the fitted model planes. The ceiling is the top part of the figure with the red model; on the right with turquoise is the frontal wall with shelves; and in green is the side wall beside the window. Note the high amount of noise and curved data in which accurate plane models are estimated by MC-LSE.

## DISCUSSION AND CONCLUSIONS

In this paper, we propose MC-LSE, an iterative method to accurately reconstruct objects composed of planes from 3D point clouds captured by cameras. The fundamental aspect of this method is the use of geometric constraints given in the object by design. These constraints provide a robust solution for estimating the planes that are able to deal with noisy data and in the presence of a high number of outliers. The optimization process with the aim of obtaining the model is performed for all planes of the object at the same time. This allows a minimization of the error as a whole while fulfilling the geometric constraints, improving partial solutions for individually fitting planes that do not guarantee a minimum error for each plane or, of course, a fitting of the model under geometric constraints.

To fit the planes to the noisy 3D data, first, MC-LSE provides an initial estimation of the regions of points captured by the 3D camera representing the planes of the model by means of an unsupervised clustering process. It is based on the $k$-means algorithm and aims to cluster at most the number of faces expected for the object. The input of the method is a point in $\mathbb{R}^6$ as a result of projecting the input point cloud onto a 6-dimensional space formed by each point and its normal vector calculated from the neighborhood. Working on this space allows the method to discriminate better points close to the edges that belong to different faces. However, for very noisy data that consider the normal vector, the clustering process can lead to errors assigned to different faces, points that are close in location. Consequently, the method considers the specificity of the application, weighting the coordinates of location and the normal vector by coefficients.

Once the initial clusters are estimated, they are the input for the core linear regression-based supervised step that learns the planes simultaneously, minimizing the residuals for each cluster under the geometric constraints. Although accurate for the clusters, the output of the regression is highly dependent on the clusters provided by the unsupervised step. Hence, the final proposed step consists of a re-assignment of all input points to the plane models calculated by regression. The re-assignment is performed by the closest plane using (again) the six-dimensional space composed of the point coordinates and the normal vector. This provides a minimization of the residuals of the previous step. In this way, the method iterates refining the solution until a threshold of points is not re-assigned. Experimental results show that the supervised regression and the reassignment step produce model results converging to a minimum in a few iterations. It allows the selection of the threshold again by considering the specificity of the application to refine the solution depending on the remaining time (*i.e.* real-time purposes).

The experiments in this study allow us to validate the method in the presence of different noise levels and points of view from an RGBD camera. Moreover, the comparison with LSE as baseline (and considered as part of many methods) and the other state-of-the art methods (RANSAC variant and MC-RANSAC) provides the big picture of the performance of the method. LSE and RSPD provide the best results according to processing time, which is too far from the obtained results for MC-LSE, but the error is extremely high for considerable noise levels. The use of this method as part of a more robust method should be considered. On the other hand, the other methods (Global-L0, CC-RANSAC, MC-RANSAC, Prog-X and Prior-MLSAC), and ours provide robust estimations of planar models based on inliers, but the computational cost is higher (according to the number of outliers, model parameters, etc.). The MC-RANSAC adds steps to consider the constraints among faces, allowing it to provide highly accurate results at the expense of dramatically increasing the computational cost. The proposed MC-LSE can provide the best results at a balanced time (comparable to the C++ implementation of Prog-X). Moreover, to validate in a real case study, a reconstruction based on a planar-based marker (cube) registration method using a Kinect sensor was presented.

Consequently, the importance and relevance of the proposal could be analyzed from the point of view of the computer vision problems that have to deal with planes as the core geometric model present everywhere in the hand-made 3D real world. As mentioned
previously, they were designed with geometric constraints that could be exploited. In this experimental setup, the validation was performed for cubes whose visible faces were at most three orthogonal planes. This is a subset of problems in which the solution can be considered, but it is widely present in many applications. For example, the case study for 3D markers, but also for mapping in robotic purposes such as SLAM, in which different views of a corridor could be seen at most as three orthogonal planes (floor, wall and ceil).

The generalization of the proposal for many orthogonal planes is direct, and strict constraints could be incorporated without changing the proposal, as we analyzed in the scene reconstruction case study. This case is frequent for non-calibrated cameras, which are not able to provide orthogonal planes even though they exist because of the geometric distortions of the camera. Hence, it is recommended that intrinsic calibration be used to minimize the effects of optical distortions on the image.

In future research, we plan to model other geometrical constraints for planes that can be useful for other applications (for example, we plan to use other geometrical objects for calibration purposes of a multi-camera system). At the same time, we would like to introduce a random selection of the landmarks of the first iteration to not depend on the first selection of points. This could be approached from a RANSAC point of view but speeding up the solution based on the characteristics of the method. This is because it could provide the hypothesis of the planar model considering all planes while reducing the number of iterations that the method should perform. Finally, the clustering-based unsupervised step could be improved by incorporating other characteristics of the scene as the Prior-MLESAC does to reduce the iterations needed to obtain the model fitting and, consequently, the processing time.

### Funding
This work was supported by the French ANR project LIVES (ANR-15-CE23-0026-03) and the Spanish State Research Agency (AEI) and the European Regional Development Fund (FEDER) under projects TIN2017-89069-R and PID2020-119144RB-I00. There was no additional external funding received for this study. The funders had no role in study design, data collection and analysis, decision to publish, or preparation of the manuscript.

### Grant Disclosures
The following grant information was disclosed by the authors:
French ANR: LIVES (ANR-15-CE23-0026-03).
Spanish State Research Agency (AEI).
European Regional Development Fund (FEDER): TIN2017-89069-R and PID2020-119144RB-I00.

### Competing Interests
The authors declare that they have no competing interests.

## Author Contributions

- Jorge Azorin-Lopez conceived and designed the experiments, performed the experiments, analyzed the data, performed the computation work, prepared figures and/or tables, authored or reviewed drafts of the paper, and approved the final draft.
- Marc Sebban conceived and designed the experiments, analyzed the data, prepared figures and/or tables, authored or reviewed drafts of the paper, and approved the final draft.
- Andres Fuster-Guillo conceived and designed the experiments, analyzed the data, authored or reviewed drafts of the paper, and approved the final draft.
- Marcelo Saval-Calvo performed the experiments, performed the computation work, authored or reviewed drafts of the paper, and approved the final draft.
- Amaury Habrard analyzed the data, authored or reviewed drafts of the paper, and approved the final draft.

## Data Availability

The data and code are available at http://tech4d.dtic.ua.es/resultados#mclsecodigo.

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
