# Peer review of "Iterative multilinear optimization for planar model fitting under geometric constraints"

_PeerJ Computer Science, doi:10.7717/peerj-cs.691_

## Round 0.1 · original submission · Major Revisions

The recommendations from the reviewers are consistent. A revision is recommended.

Reviewer 1 ·

Basic reporting

1. The literature review focused on three principal categories, but there were few related research papers from 2018-2020. I suggest that the manuscript include state-of-the-arts to help readers understand the latest developments in this field.

2. In line 83, “Region growing methods are robust to noisy point clouds” should be explained in detail. Generally speaking, region growing methods suffer from noise.

Experimental design

1. In table 2-3, with the increase of noise, MC-LSE gradually loses its advantage in Dist. Error Cluster Error, but which didn’t influence the algorithm accuracy in terms of angle error and model error. The author needs to explain it in the paper. In addition, the positions of Tables 2 and 3 may need to be switched. Because you mentioned Table 3 first in the article instead of Table 2.

2. In table4,“Table 4. Results (mean ± standard deviation) of the pre-clustering method considering different input data with respect to various levels of Gaussian noise (from 1.E-05 to 1.E-04) and viewpoints”, but there is no Gaussian noise information in the table.

Validity of the findings

Although the results showed that the proposed method was better, the methods for comparisons were old methods. Comparisons with state-of-the-art methods are necessary.

Additional comments

The manuscript presented a planar fitting model based on the least-squares estimation method (LEM). Different from the ordinary LEM, the model was robust to noises by involving the geometric constraints. The method is simple but interesting, and the article is well organized. In addition to a few comments, I encourage the authors to check the mistakes in their manuscript to enhance the levels of paper presentation. For example,
In line42, “Computer vision plays an important role (in) providing methods for…”
In line 135, “It allow (allows) us to provide a high accuracy…”
In line 136, “with respect (to) other iterative methods…”
In line 464, “edges that belongs (belong) to different faces.”

Reviewer 2 ·

Basic reporting

Some formatting problems will be improved such as formula annotation, table header.

Experimental design

no comment

Validity of the findings

no comment

Additional comments

This paper proposed an approach to simultaneously fit different planes in a point cloud using linear regression estimators and normal vectors to the planes with three steps: a clustering-based unsupervised step, regression-based supervised step and a reassignment step. Experimental results show the effectiveness of the proposed method. It is a nice job of providing good accuracy/speed trade-off in the presence of noise and outliers.

Reviewer 3 ·

Basic reporting

The English need some polishing in terms of sentence construction.
Literature references are appropriate.
Figures and tables of acceptable quality.
The manuscript is self-contained with relevant results to hypotheses.

Experimental design

The experimental design appears to be original.
The research question is well defined, relevant and meaningful.
A rigorous investigation has been carried out.
The methods have been described in sufficient details.

Validity of the findings

The method appears to be novel.
Data have been provided and appears to be statistically sound.
Conclusions are well stated.

Additional comments

The English should be checked and few bad sentences here and there should be corrected which will make the manuscript more attractive.

Annotated reviews are not available for download in order to protect the identity of reviewers who chose to remain anonymous.

---

## Round 0.2 · accepted · Accept

The paper has been improved following the comments, therefore, the paper is ready to be accepted.

Reviewer 1 ·

Basic reporting

no comment

Experimental design

no comment

Validity of the findings

no comment

Additional comments

no comment